# Modular one-pot assembly of CRISPR arrays enables library generation and reveals factors influencing crRNA biogenesis

Chunyu Liao[1,2], Fani Ttofali[1], Rebecca A. Slotkowski[1], Steven R. Denny[1], Taylor D. Cecil[1], Ryan T. Leenay[1], Albert J. Keung[1] & Chase L. Beisel [1,2,3]

CRISPR-Cas systems inherently multiplex through CRISPR arrays—whether to defend against different invaders or mediate multi-target editing, regulation, imaging, or sensing. However, arrays remain difficult to generate due to their reoccurring repeat sequences. Here, we report a modular, one-pot scheme called CRATES to construct CRISPR arrays and array libraries. CRATES allows assembly of repeat-spacer subunits using defined assembly junctions within the trimmed portion of spacers. Using CRATES, we construct arrays for the single-effector nucleases Cas9, Cas12a, and Cas13a that mediated multiplexed DNA/RNA cleavage and gene regulation in cell-free systems, bacteria, and yeast. CRATES further allows the one-pot construction of array libraries and composite arrays utilized by multiple Cas nucleases. Finally, array characterization reveals processing of extraneous CRISPR RNAs from Cas12a terminal repeats and sequence- and context-dependent loss of RNA-directed nuclease activity via global RNA structure formation. CRATES thus can facilitate diverse multiplexing applications and help identify factors impacting crRNA biogenesis.

---

[1] Department of Chemical and Biomolecular Engineering, North Carolina State University, Raleigh, NC 27695, USA. [2] Helmholtz Institute for RNA-based Infection Research, Josef-Schneider-Straße 2/D15, 97080 Würzburg, Germany. [3] Medical Faculty, University of Würzburg, 97080 Würzburg, Germany. Correspondence and requests for materials should be addressed to C.L.B. (email: chase.beisel@helmholtz-hiri.de)

CRISPR-Cas systems represent RNA-directed immune systems whose programmable nucleases have become powerful tools for genome editing, gene regulation, imaging, and diagnostics[1,2]. One unique feature of the immune systems is their inherent ability to multiplex through CRISPR arrays. Arrays comprise alternating conserved ~36-bp repeats and invader-derived ~30-bp spacers that are transcribed into precursor CRISPR RNAs (pre-crRNAs) and processed into individual CRISPR RNAs (crRNAs) (also generally called guide RNAs). Each crRNA then directs the nuclease to bind and cleave nucleic-acid targets complementary to the spacer and flanked by short defined sequences such as protospacer-adjacent motifs (PAMs)[3]. As each crRNA would direct the nuclease to a distinct target, a single CRISPR array can orchestrate the simultaneous targeting of multiple sequences.

The multiplexing capacity of CRISPR arrays was well recognized before the advent of CRISPR technologies[4,5], yet multiplexing efforts originally revolved around engineered single-guide RNAs (sgRNAs) utilized by the Type II Cas9 nuclease[6]. These sgRNAs circumvented the need for either a tracrRNA or RNase III that both participate in crRNA biogenesis[7], thereby allowing DNA targeting with only an sgRNA and Cas9. However, because an sgRNA represents the final version of the crRNA, multiplexing with a single DNA construct has instead required workarounds such as expressing individual sgRNAs from separate promoters or joining sgRNAs with intervening cleavage domains[8–11]. The more recent discovery that Type V Cas12a and Type VI Cas13 single-effector nucleases can process CRISPR arrays without accessory factors[12–15] renewed interest in synthetic CRISPR arrays and led to recent examples of using arrays for multiplexed genome editing and gene regulation in bacteria and eukaryotes[16–19]. CRISPR arrays further offer notable advantages over sgRNA arrays, including compatibility with any CRISPR nuclease and repeat-spacer subunits that are much smaller (~66 bp) than those typically used in sgRNA arrays (~150–400 bp). CRISPR arrays thus hold widespread potential for multiplexing applications, including enhanced gene drives[20], coordinate regulation of metabolic pathways[21], predictable gene deletions[22], proximal CRISPR targeting[23], multi-pathogen antimicrobials[24], multiplexed base editing[25], combinatorial nucleic-acid sensing[18,26], and combinatorial screens[27–29].

Despite their desirable features, CRISPR arrays remain to be widely adopted. A prevailing reason has been the challenge of efficiently generating arrays. For instance, groups could request commercial vendors to chemically synthesize arrays as linear double-stranded DNA (dsDNA) gene fragments, although the reoccuring repeats inherent to the arrays currently interfere with fragment construction. Separately, many established assembly techniques have been developed for sgRNA arrays, with Golden-Gate assembly the most common[8,9], although none of the techniques have been extended to CRISPR arrays. Instead, the few available techniques for generating CRISPR arrays involve annealing shorter oligonucleotides into repeat-spacer subunits and using overhangs at the ends of each subunit to form the intervening assembly junctions (Table S1)[16,17,30–32]. While these techniques have been used to assemble a handful of individual arrays harboring two to four spacers, they require multiple assembly steps or rely on overhangs derived from the existing repeat or spacer sequences. The lack of flexibility in selecting assembly junctions in particular heavily restrains the generation of large arrays or array libraries. What is needed is a strategy for assembling CRISPR arrays that can be performed in one pot and decouples the overhangs from the selected repeat and spacer sequences.

Here, we address this challenge through the development of CRATES (CRISPR Assembly through Trimmed Ends of Spacers),

a modular, one-pot assembly scheme for CRISPR arrays. CRATES takes advantage of the portion of crRNA spacers that does not participate in target recognition and often undergoes trimming as part of crRNA biogenesis for single-effector Cas nucleases[7,15,33]. By introducing a defined assembly junction into this region, compatible overhangs can be selected that are fully decoupled from the targeting portion of the spacer and the conserved repeat. We show that this strategy allowed the efficient construction of larger arrays, a 125-member library of three-spacer arrays, and composite arrays utilized by multiple nucleases. We also use the assembled arrays to elucidate factors relevant to the design of synthetic arrays and the evolution of naturally occurring arrays, including native terminal repeats in Cas12a arrays preventing the generation of an extraneous yet functional crRNA, and the contribution of global secondary structure to crRNA abundance and crRNA-directed nuclease activity. In total, CRATES is expected to streamline multiplexing with numerous CRISPR single-effector nucleases and facilitate the interrogation of the processing, function, and evolution of CRISPR arrays.

## Results

**CRATES is a one-pot assembly scheme for CRISPR arrays**. We began with the Cas12a nuclease from *Francisella novicida* U112 (FnCas12a), whose crRNA spacers are known to undergo trimming at its 3′ end from 30 nts in the transcribed array to ~23 nts in the processed crRNA (Fig. 1a)[15]. Furthermore, recent crystal structures of the FnCas12a:crRNA ribonucleoprotein complex bound to target DNA showed that only the first 20 nts of the guide portion of the crRNA participated in target hybridization[34]. These insights led us to hypothesize that overhangs for the assembly of repeat-spacer subunits could be placed at the 3′ end of each spacer without interfering with crRNA-directed DNA targeting—a hypothesis we confirmed based on similar plasmid clearance activities in *E. coli* with or without a junction sequence (Supplementary Fig. 1). The junctions used to assemble repeat-spacer subunits parallel those used for the assembly of multiple DNA fragments by Golden-Gate Assembly[35] with one notable difference: rather than relying on overhangs generated using Type IIS restriction enzymes, we can create junctions comprising 5′ or 3′ overhangs of any length or sequence by annealing two short, ~66-nt oligonucleotides to form each repeat-spacer subunit. We term the resulting assembly scheme CRATES for CRISPR Assembly through Trimmed Ends of Spacers.

We first designed a base construct for the assembly and expression of FnCas12a arrays in the bacterium *Escherichia coli* (Fig. 1b). The construct contained a total of six components: a constitutive promoter to drive transcription of the array, two Type IIS restriction sites for inserting multiple repeat-spacer subunits at one time, a GFP reporter construct that is excised as part of array assembly, a 3′ repeat so the final array begins and ends with repeats, and a terminator to halt transcription of the array. Repeat-spacer subunits were designed to contain 5′ and/or 3′ overhangs with highly dissimilar sequences previously validated for efficient modular assembly[36], thereby diminishing the frequency of misassembly. We chose 4-nt overhangs given their use with assembly techniques based on Type IIS restriction enzymes, although other overhang lengths could be tested. The repeat-spacer subunits and the base construct were then assembled into an array in a one-pot reaction that cycles between digestion of the backbone with a Type IIS restriction enzyme and ligation by T4 DNA ligase in a thermocycler. Figure 1b depicts the assembly of a three-spacer array with four distinct junctions.

**CRATES enabled efficient assembly of arrays and libraries**. We first explored how well CRATES could be used to assemble

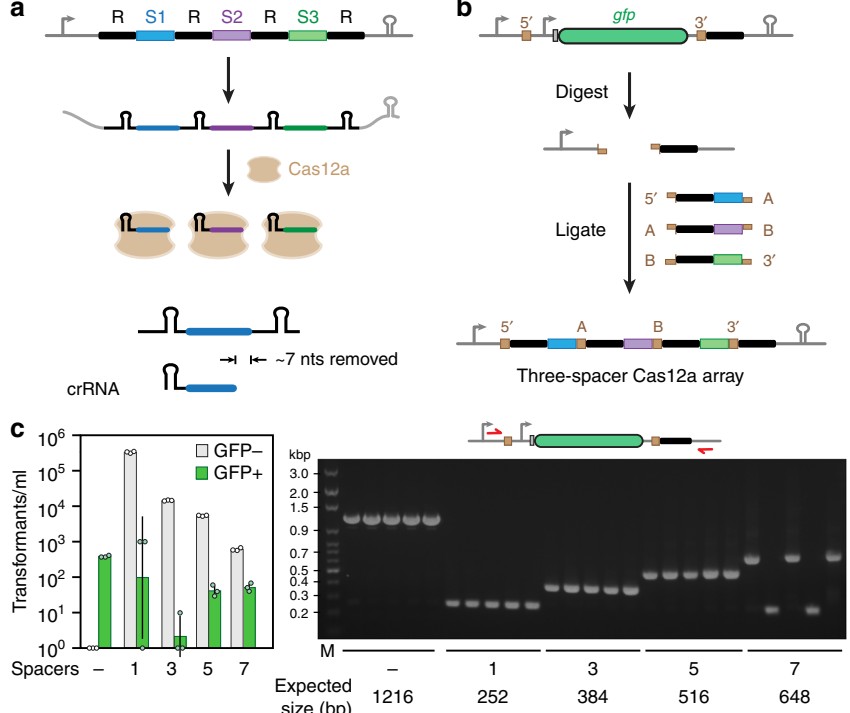

**Fig. 1** A modular, one-pot assembly scheme for CRISPR-Cas12a arrays based on spacer trimming. **a** Processing of the transcribed CRISPR array by the Cas12a nuclease. Each array comprises alternating repeats (R, black bars) and spacers (S1–S3, colored bars). Processing includes 3′ trimming of the spacer to form the final crRNA. **b** Cloning scheme for the assembly of multi-spacer arrays recognized by Cas12a. A GFP-dropout construct is flanked by two Type IIS restriction sites and a 3′ direct repeat. The digested construct is ligated with annealed oligonucleotides encoding individual repeat-spacers in one step. The 3′ repeat is located in the backbone plasmid so the insertion of repeat-spacer subunits results in an array that begins and ends with a repeat. The sequence and 5′ or 3′ directionality of the junction overhangs determine the order of assembly. The resulting assembly junctions fall within the trimmed portion of the processed guide RNA and therefore would not be involved in target recognition. **c** Efficient assembly of up to a seven-spacer array. Assembly efficiency was based on the relative proportion of fluorescent or non-fluorescent *E. coli* colonies (left) and the correct insert size of non-fluorescent colonies subjected to colony PCR (right). Values represent the geometric mean and S.D. from three independent transformations starting from separate colonies. See Supplementary Table 3 for the specific sequences used. The dash represents the original GFP-dropout construct. Source data are provided as a Source Data file

individual arrays containing different numbers of spacers. We designed arrays recognized by FnCas12a with up to seven spacers containing distinct 26-nt sequences and 4-nt junctions. Following the assembly reaction and transformation into *E. coli*, we counted the relative number of fluorescent or non-fluorescent cells, where non-fluorescent cells lost the GFP expression construct and therefore would be expected to contain assembled arrays (Fig. 1c). We found that the total number of transformants decreased for arrays with more spacers, in line with lower efficiencies when assembling more fragments at one time. However, non-fluorescent colonies always outnumbered fluorescent colonies by a factor between 30 and over 1000. By contrast, only fluorescent colonies were observed in the absence of any added repeat-spacer subunits. Colony PCR of five random, non-fluorescent colonies yielded the correct band size 100% of the time (5/5) for arrays containing up to five spacers and 60% of the time (3/5) for arrays with seven spacers (Fig. 1c). Colony PCR of individual colonies yielded the expected band size, while Sanger sequencing confirmed that the arrays contained the expected sequence. In the case of the two negative clones, the smaller PCR products were in line with formation of arrays with a single spacer. CRATES therefore represents a simple and efficient approach to assemble large CRISPR arrays up to and potentially exceeding seven spacers.

Given that the junction sequences are decoupled from the targeting portion of the spacers, we asked how well CRATES could be used to generate a library of CRISPR arrays. As a proof-

of-principle demonstration, we designed three-spacer FnCas12a arrays with five unique spacers that could occupy each location in an array, yielding a total of 125 different arrays (Fig. 2a). After performing the assembly reaction in two independent experiments, we plated 10 μl of each batch of recovering transformants. Out of the 251 resulting colonies between the two runs, only one colony was green (Fig. 2b). Furthermore, colony PCR screening of random white colonies showed that 95% (18/19) of the screened colonies harbored an assembled array (Fig. 2b). Finally, we performed next-generation sequencing to determine the identity and abundance of the arrays in one of the libraries. We found that all arrays were present at relatively uniform abundance, with a 7.4-fold ratio between the highest-abundance and lowest-abundance arrays (Fig. 2c). Deeper analysis of the relative array abundances revealed that the slight variability in abundance can be mostly attributed to a lower stoichiometry or cloning efficiency of two spacers in the 3′ location (S3k, S3o), suggesting that adjusting the levels of the corresponding repeat-spacer subunits could further narrow the distribution of array abundances in the library (Fig. 2d). These results therefore show that CRATES can be used to create libraries of CRISPR arrays.

**CRATES enabled assembly of arrays for different Cas nucleases.** Given the success generating individual CRISPR arrays as well as array libraries, we next evaluated whether the resulting arrays could guide multiplexed DNA targeting. In particular,

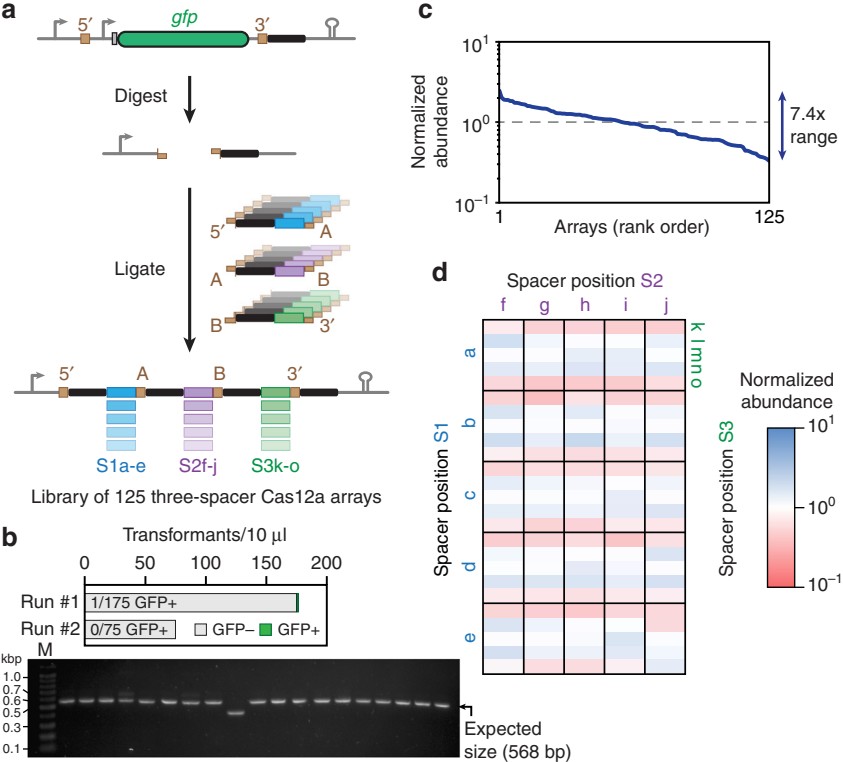

**Fig. 2** CRATES can be used to efficiently generate a library of assembled CRISPR arrays in one pot. **a** Cloning scheme for the CRISPR array library. For this demonstration, a three-spacer array was generated, where each spacer could be one of five unique sequences. The resulting library contains 125 members. Note that the protocol for CRATES was slightly modified to further improve the cloning efficiency—see Methods. **b** Two independent libraries were generated followed by counting the number of green and white colonies. Nineteen white colonies from run #2 were then screened by colony PCR. All but one colony gave the expected band size of 528 bp. **c** Distribution of array abundances within the library from run #2. The distribution was assessed by next-generation sequencing. **d** Abundances of individual arrays within the 125-member library. The relative abundances are derived from next-generation sequencing results. A value of 1 represents the expected abundance if every library member is equally abundant. Source data are provided as a Source Data file

the advantage of CRATES over prior assembly techniques (Supplementary Table 1) is that the junction sequences are decoupled from the repeat sequence, allowing us to readily use other repeat sequences when constructing arrays. We began with the Cas12a nuclease from *Acidaminococcus* sp. BV3L6 (AsCas12a), a homolog of FnCas12a exhibiting robust editing activity in mammalian cells[15]. We evaluated crRNA-directed nuclease activity based on plasmid clearance in *E. coli* (Fig. 3a). As part of the assay, a plasmid with a targeting or no-spacer CRISPR array was transformed into cells harboring the AsCas12a plasmid and a plasmid containing the target sequence. Successful plasmid clearance resulted in a large reduction in the number of antibiotic-resistant colonies for the targeting array versus the no-spacer array. We designed three spacers against three distinct target sequences. CRATES was then used to construct either single-spacer arrays or a three-spacer array using the conserved repeat sequence associated with AsCas12a. Performing the plasmid clearance assay, we found that the single-spacer arrays cleared only their cognate target plasmid, while the array with all three spacers cleared all target plasmids (Fig. 3a and Supplementary Fig. 2A).

As a further extension of multiplexing with the Cas12a nuclease, we explored multiplexed gene regulation with a catalytically dead version of FnCas12a (dFnCas12a). Three RuvC domains have been implicated in DNA cleavage, where single point mutations completely or partially disrupted cleavage in vitro in the first two domains (D917A, E1006A) or the third domain (D1255A), respectively[12,15]. We initially tested different

combinations of the impact of thse mutations on FnCas12a using the plasmid clearance assay in *E. coli* (Supplementary Fig. 3A). All mutations disrupted plasmid clearance, indicating that mutating even the third RuvC domain of FnCas12a could disrupt cleavage in vivo. We then measured gene repression in *E. coli* by expressing a single-spacer array targeting the *lacZ* promoter upstream of *gfp* on a plasmid (Supplementary Fig. 3A). Interestingly, flow cytometry analysis revealed that, while all combinations of mutations led to a reduction in GFP levels compared to a non-targeting control, the E1006A mutant exhibited only partial repression activity in line with a similar observation reported recently[37]. Furthermore, we achieved multiplexed gene repression in *E. coli* with the triple mutant (D917A, E1006A, D1255A) when targeting three different promoters (*araB*, *lacZ*, *lacIq*) controlling *gfp* on different plasmids (Supplementary Fig. 3B). To extend these findings to eukaryotic cells, we employed the FnCas12a double mutant (D917A, E1006A) fused to the VP64 activation domain in the budding yeast *Saccharomyces cerevisiae* in line with previous work[38] to assess multiplexed targeting of a CYC1 promoter upstream of a chromosomal copy of *yegfp* (Supplementary Fig. 3C). In line with the synergistic impact of recruiting multiple copies of VP64 to the same promoter[39], flow cytometry analysis showed that the three-spacer arrays consistently yielded more yEGFP-expressing cells with higher fluorescence than those targeted by single-spacer arrays, but only in the presence of VP64 (Supplementary Fig. 3C).

Next, we asked if CRATES could be extended to generate arrays utilized by other CRISPR single-effector nucleases, given

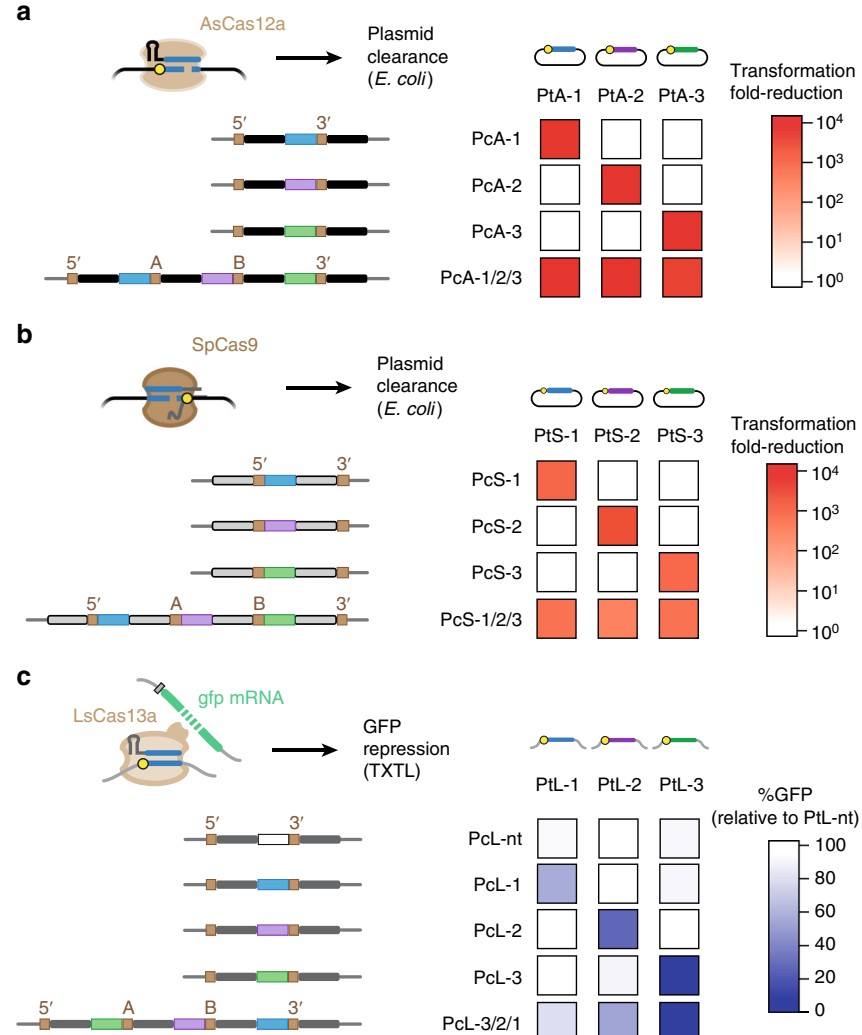

**Fig. 3** CRATES can be used to generate CRISPR arrays that function with different Cas single-effector nucleases. **a** Multiplexed plasmid clearance by AsCas12a in *E. coli*. Spacers were designed to target a distinct protospacer flanked by a PAM (yellow circle) in a transformed plasmid. The assembly junction is located at the 3′ end of the spacer to match the location of spacer trimming. *E. coli* cells harboring the AsCas12a plasmid and the target plasmid were transformed with a plasmid encoding the indicated CRISPR array or a no-spacer array, and the fold-change in the number of transformants was quantified. Values represent the average of three independent transformation experiments starting from separate colonies. **b** Multiplexed plasmid clearance by SpCas9 in *E. coli*. See **a** for details. The Cas9 plasmid also encodes the tracrRNA. The assembly junction is located at the 5′ end of the spacer to match the location of spacer trimming. Values represent the average of three independent transformation experiments starting from separate colonies. **c** Multiplexed RNA sensing by LsCas13a in a cell-free transcription–translation system. Reactions were conducted for 16 h following the addition of the LsCas13a plasmid, the indicated array plasmid, a plasmid expressing one of the targets, and the GFP plasmid. Target recognition leads to non-specific degradation of the GFP mRNA by LsCas13a, thereby reducing GFP production. GFP levels from end-point fluorescence measurements are reported relative to that for a non-targeted transcript (PtL-nt) and the same targeting array. Values represent the average of two TXTL experiments and are representative of at least three experiments conducted on different days. See Supplementary Fig. 5 for the assay and GFP time courses. All arrays were assembled using the junctions specified in Supplementary Table 4. Source data are provided as a Source Data file

that the associated crRNA spacers also undergo trimming and do not rely on the full spacer for target recognition[7,33]. We first began with the Type II Cas9 nuclease from *Streptococcus pyogenes* (SpCas9), where the associated crRNA spacers are trimmed by ~10 nts from the 5′ end and only 20 nts are necessary for DNA targeting[6,7]. To account for the location of the guide region in the spacer utilized by the nuclease, we modified the base assembly construct by placing the repeat upstream rather than downstream of the GFP-dropout cassette. We also encoded the tracrRNA required for crRNA processing in the Cas9 expression plasmid, and we relied on the RNase III endogenous to *E. coli*. As expected, the plasmid clearance assay resulted in the single-spacer arrays only clearing their cognate target plasmid and the three-spacer

assay clearing each of the three target plasmids (Fig. 3b and Supplementary Fig. 2B). We then transitioned to the Type VI single-effector nuclease Cas13a from *Leptotrichia shahii* (LsCas13a). This nuclease targets RNA, where target recognition leads to LsCas13a non-specifically degrading other RNAs in the vicinity[40]. As the crRNA spacer for LsCas13a is trimmed on the 3′ end[13,33], we developed a base assembly construct similar to that for Cas12a. We then devised an RNA-sensing assay using an *E. coli* cell-free transcription–translation (TXTL) system[41], where DNA encoding LsCas13a and the constructed CRISPR array were mixed with DNA expressing a target RNA and a non-targeted GFP reporter (Supplementary Fig. 4A, B). The GFP transcript would then undergo non-specific degradation only when the

processed crRNA is paired with its target, resulting in curtailed production of GFP. As expected, GFP expression in the RNA-sensing assay was reduced compared to a no-spacer array only when the spacer and its cognate target were both present (Fig. 3c and Supplementary Figs. 2C and 4C). Furthermore, a three-spacer array reduced GFP in the presence of any of the three target transcripts but not of a non-targeted transcript, resulting in the simultaneous sensing of multiple RNA species.

**CRATES-assembled composite arrays mediated coordinated targeting**. Emerging examples have shown that orthogonal Cas nucleases can be combined to simultaneously perform different CRISPR functions[23,42–44]. However, in each case, the guide RNAs had to be transcribed from separate expression constructs. We therefore asked if CRATES could be used to generate individual CRISPR arrays that are transcribed and processed into crRNAs recognized by multiple Cas nucleases--what we term composite arrays. Fig. 4a illustrates a composite array composed of a two-spacer SpCas9 array followed by a two-spacer FnCas12a array, while Fig. 4b shows how CRATES could be readily adopted to clone the composite array in one pot.

We first assessed the ability of these arrays to coordinate plasmid clearance by SpCas9 and gene repression by the dFnCas12a triple mutant in *E. coli* (Fig. 4c). As part of the assessment, we designed two spacers for each nuclease targeting the *lacZ* or *lacIq* promoter controlling *gfp* in the reporter plasmid. We then constructed arrays containing each spacer or all four spacers in two configurations (Fig. 4c). *E. coli* cells harboring the SpCas9/tracrRNA or dFnCas12a plasmid and either GFP reporter plasmid were transformed with each array plasmid, and the transformation efficiency (with SpCas9) or GFP fluorescence (with dFnCas12a) was then assessed in comparison to the no-spacer array plasmid. Similar to Fig. 3b, we relied on RNase III endogenous to *E. coli* for processing of the portion of the arrays used by SpCas9. As expected, the single-spacer arrays yielded plasmid clearance or GFP repression when matched with their nuclease and targeted plasmid, while one of the four-spacer composite arrays (Pc7/8/9/10) yielded plasmid clearance or GFP repression comparable to that of all single-spacer arrays (Fig. 4c and Supplementary Fig. 2D). Interestingly, the other four-spacer composite array with swapped SpCas9 spacers (Pc8/7/9/10) exhibited greatly reduced plasmid clearance of one target plasmid by SpCas9, suggesting that the sequence and context of the spacers in an array can impact the resulting crRNA-directed nuclease activity.

As a separate demonstration, we asked if composite arrays could be used to direct the blocking of off-target sites with a catalytically dead nuclease as part of genome editing with a separate, catalytically active nuclease. We chose SpyCas9 for on-target cleavage and the FnCas12a triple mutant to block off-target sites (Fig. 4d). We further selected one of the original examples of off-target cleavage by SpCas9 in human cells, with an on-target site in WAS CR-4 (P-on) and two known off-target sites in STK25 (P-off1) and GNHR2 (P-off2) (Supplementary Fig. 5A)[45]. The target sites for dFnCas12a were chosen so the R-loop would extend through the NGG PAM recognized by SpCas9, thereby presumably preventing DNA recognition. Each site was further cloned upstream of the GFP reporter construct, and TXTL was used as a rapid means to assess blocking of the off-target sites but not the on-target site by FnCas12a. We also used CRATES to assemble a variant of the composite array in which the targeting SpCas9 sgRNA was placed upstream of a two-spacer FnCas12a array (Psg/b1/b2 and Psg/b2/b1). To perform the assay, we first added the dFnCas12a plasmid and a reporter plasmid along with either one plasmid encoding a composite array or two plasmids

each encoding the sgRNA and a single-spacer FnCas12a array (Supplementary Fig. 5C). We then added the plasmid encoding SpCas9 and measured GFP fluorescence over time (Supplementary Fig. 5D). The protection efficiency was then calculated based on the rate of GFP production in comparison to that when expressing a no-spacer FnCas12a array and the targeting sgRNA (0% protection) or a non-targeting sgRNA (100% protection). We found that both tested composite arrays inhibited cleavage by SpCas9 at both off-target sites at efficiencies similar to those when expressing the SpCas9 sgRNA and single-spacer FnCas12a arrays separately (Fig. 4e and Supplementary Fig. 2E). Critically, there was minimal protection of the on-target site. These results suggest that composite arrays can be used to block off-target sites and improve on-target specificity by Cas nucleases, although more testing is needed beyond this cell-free demonstration.

**Secondary structure contributes to crRNA biogenesis**. Given the ease in constructing CRISPR arrays using CRATES, we asked if the resulting arrays could shed light on crRNA bio-genesis and the resulting crRNA-directed nuclease activity. We began with the synthetic, seven-spacer array for FnCas12a that we constructed (Fig. 1c) and asked how it undergoes processing by FnCas12a when transcribed. We used two different expression systems, TXTL and transient transfection in HEK293T cells, to co-express the array and FnCas12a and then performed RNA-seq analysis on the purified small RNAs[41]. In both cases, next-generation sequencing showed that the transcribed array was processed into ~44-nt crRNAs similar to prior work[12,15] (Fig. 5). We also noted that a portion of the crRNAs included some of the junctions, potentially affecting targeting if this region is involved in target hybridization for other CRISPR nucleases.

The RNA-seq analysis further revealed two striking observations: widely varying crRNA abundances and a crRNA being derived from the 3′ repeat and the downstream sequence. The first observation parallels RNA-seq analyses of CRISPR arrays in their natural contexts (Fig. 5)[7,15,33,46]. Interestingly, the relative abundances trended similarly between TXTL and HEK293T cells, suggesting that the underlying mechanism was independent of the cell expression system. An ensuing question is what explains the variability in crRNA abundance. Given the importance of a characteristic hairpin within the repeat as part of crRNA processing by Cas12a[15], we hypothesized that repeats in highly structured regions lacking a formed hairpin would yield poorly abundant crRNAs. To explore this possibility, we predicted the secondary structure of the transcribed, seven-spacer FnCas12a array using NUPACK (Supplementary Fig. 6A, B)[47]. In line with our hypothesis, the least abundant crRNAs (S2, S3, S4, S5) were part of highly structured regions that often disrupted the characteristic hairpin recognized by FnCas12a in each repeat. By contrast, the most-abundant crRNAs (S1, S6, S7), were generally unstructured and formed the hairpin in each repeat. Similar trends were observed for RNA-seq analysis of the native nine-spacer array from *F. novicida* U112 (Supplementary Fig. 6C, D). Spacer S7 in this native array appeared to be an exception because the hairpin of the repeat is correctly formed and the spacer is mostly unstructured; however, the prediction of a 10-bp hairpin through the 3′ end of the spacer would be expected to inhibit processing based on our recent work[48]. These results therefore suggest that the global secondary structure of a transcribed array can help explain the variable abundance of the individual crRNAs for both natural and synthetic arrays. There were some differences in the final profile of crRNAs generated in TXTL versus HEK293T cells (e.g., different extents of processing), which likely reflects the associated cohorts of

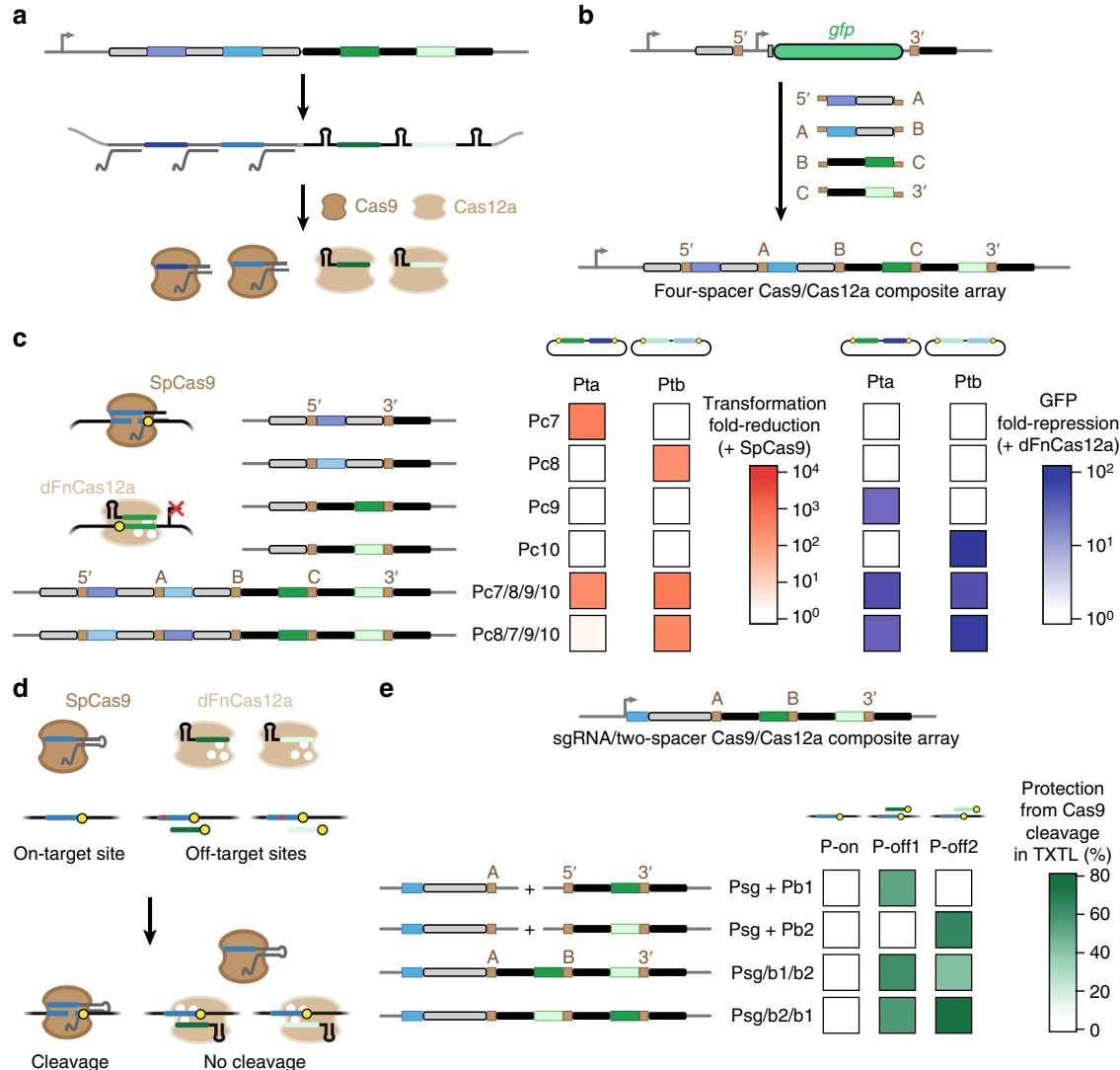

**Fig. 4** Composite arrays allow multi-nuclease targeting from a single transcript. **a** Processing of a composite array encoding spacers used by Cas9 and Cas12a. **b** One-step assembly of composite arrays. An array encoding two CRISPR-Cas9 spacers and two CRISPR-Cas12a spacers is shown. The GFP-dropout construct contains an upstream SpCas9 repeat and a downstream FnCas12a repeat to accommodate the orientations of the assembly junctions. **c** Coordinated plasmid clearance by SpCas9 and gene repression by the dFnCas12a triple mutant (D917A, E1006A, D1255A) in *E. coli*. To assess plasmid clearance, cells harboring the SpCas9 plasmid and a target plasmid were transformed with a plasmid encoding the indicated CRISPR array or a no-spacer array. To investigate the GFP repression, cells harboring the dFnCas12a plasmid, a target plasmid, and a plasmid encoding the indicated CRISPR array or a no-spacer array were assessed by flow cytometry analysis. Values represent the average of three independent experiments starting from separate colonies. See Supplementary Tables 3 and 4 for more information on the target constructs. **d** Enhancing the specificity of DNA cleavage by SpCas9 by blocking off-target sites with the dFnCas12a triple mutant. Binding of a known off-target location by dFnCas12a would block Cas9 from accessing this site, thereby reducing unintended cleavage at this site. **e** Evaluating DNA cleavage in a model cell-free TXTL system using a composite array composed of one SpCas9 sgRNA and a two-spacer FnCpf1 array. Protection from cleavage was calculated based on the relative rates of GFP production compared to target and non-targeting controls. Values represent the average of three independent TXTL experiments conducted on separate days. See Supplementary Fig. 5 for details of the target sites and a representative set of GFP time-course measurements. Source data are provided as a Source Data file

active ribonucleases as well as cell-free versus transient transient expression.

**crRNA activity can be sequence- and context-dependent.** Low-abundance crRNAs may yield reduced nuclease activity. We encountered a strong candidate when performing plasmid clearance assays in *E. coli* using three-spacer arrays with FnCas12a (Fig. 6a). We assessed all six permutations of a three-spacer array targeting three separate plasmids. Surprisingly, one out of the six arrays (PcF-2/3/1) yielded minimal clearance of the target plasmid with the *lacZ* promoter (PtF-1) in comparison to a no-spacer

array (Fig. 6a and Supplementary Fig. 2F). The robust clearance of PtF-1 by the other five arrays suggested that the sequence and context of a given spacer in an array can affect the extent of nuclease activity by the associated crRNA.

Based on our insights into varying crRNA abundance (Fig. 5), we evaluated the relative abundance of the crRNAs derived from the array with poor clearance by spacer S1 (PcF-2/3/1). We also evaluated an array in which spacer S1 and a different spacer were switched (PcF-1/3/2) to distinguish between the contributions of the spacer and the spacer position. RNA-seq analysis of the arrays co-expressed with FnCas12a in TXTL revealed that the crRNA derived from spacer S1 in PcF-2/3/1 had the lowest abundance

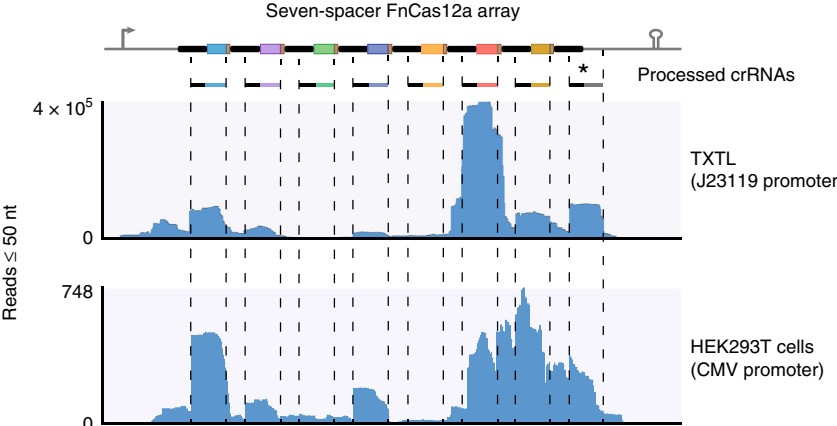

**Fig. 5** RNA-seq analysis revealed ranging crRNA abundances and an extraneous crRNA. RNA-seq analysis of the seven-spacer array from Fig. 1c co-expressed with FnCas12a in TXTL (top) or in HEK293T cells (bottom). The assembled array and FnCas12a were expressed followed by small-RNA isolation and RNA-seq analysis. Reads of no more than 50 nts were mapped to the expression construct for the synthetic FnCas12a array with seven spacers. The asterisk indicates the extraneous crRNA derived from the terminal repeat. For the experiments in HEK293T cells, the array was expressed from a CMV promoter

across the two arrays and was ~4-fold lower than the same crRNA in PcF-1/3/2 relative to the middle spacer in each array--all in line with poor plasmid clearance activity. As further confirmation, we performed Northern blotting analysis to detect the crRNA derived from spacer S1 in *E. coli* expressing FnCas12a and the arrays from PcF-2/3/1 or PcF-3/2/1, where the first two spacers were switched. This analysis showed that the crRNA derived from spacer S1 was less abundant for PcF-2/3/1 than for PcF-3/2/1 (Fig. 6c and Supplementary Fig. 7). We also noticed that the most-abundant crRNAs in the RNA-seq analyses were derived from the 3′ repeat. While these crRNAs could be titrating out available FnCas12a protein, removing the terminal repeat did not significantly change plasmid clearance ($p = 0.48$ for two-tailed $t$-test, $n = 3$) (Supplementary Fig. 8). Therefore, other factors likely account for the context-dependent loss of crRNA-directed nuclease activity.

Our analysis of the native and synthetic FnCas12a arrays suggested that the global secondary structure of the transcribed array impacted crRNA abundance, which may help explain the loss of plasmid clearance activity. We therefore predicted the secondary structure of the transcribed array using NUPACK. We found that only the array from PcF-2/3/1 formed a stable, imperfect hairpin bridging spacers S3 and S1 (base-pairing probabilities of ~0.3–1.0) that disrupted the characteristic hairpin in the intervening repeat (Fig. 6d and Supplementary Fig. 9). To evaluate whether the imperfect hairpin predicted to form within the transcribed array was responsible for poor clearance of the PtF-1 target plasmid, we introduced five different mutations into spacer S3 to destabilize the hairpin (Fig. 6d). We then repeated the plasmid clearance assays using variants of PtF-1 that matched the mutated spacers (Fig. 6e). In all cases, the mutations resulted in significantly improved plasmid clearance compared to the parental array ($p = 0.001$–$0.053$ for two-tailed $t$-test, $n = 3$), with four of the mutants fully restoring clearance activity. As a further confirmation, we introduced additional mutations to two of the mutants (m4, m5) predicted to regain a stable, imperfect hairpin (Fig. 6d). For both of the resulting mutants (m4′, m5′), plasmid clearance was significantly disrupted compared to the corresponding mutant ($p = 0.002$, $0.0042$ for two-tailed $t$-test, n = 3) (Fig. 6e). These findings strongly indicate that secondary structure formation within transcribed CRISPR arrays can interfere with crRNA processing and the ensuing cRNA-directed nuclease activity.

**The 3′ repeat of FnCas12a arrays yields an extraneous crRNA.** The second striking observation from the RNA-seq analyses was a crRNA derived from the 3′ repeat and the immediately downstream sequence outside of the array. This crRNA was observed for the seven-spacer array expressed in TXTL and HEK293T cells (Fig. 5), as well as for the two three-spacer arrays expressed in TXTL (Fig. 6b). Aside from the complications of creating an errant crRNA that could titrate available nuclease, this phenomenon could also lead to unintended DNA targeting. Given that these issues would impact native CRISPR-Cas systems, natural terminal repeats in Cas12a arrays may be under selective pressure to accumulate mutations that prevent processing. Correspondingly, analysis of diverse terminal repeats from Type V-A CRISPR-Cas systems in 14 different bacteria revealed that 79% (11/14) of the native terminal repeats harbored mutations from the array's consensus repeat sequence expected to disrupt the sequence or predicted secondary structure in regions critical for Cas12a recognition (Fig. 7a)[12,15]. As further support, prior RNA-seq analysis of the transcribed CRISPR array native to *F. novicida* U112 showed no obvious extraneous crRNA (Supplementary Fig. 6C)[15]. We also found that expressing the seven-spacer array with the native terminal repeat from *F. novicida* in HEK293T cells eliminated the full-length extraneous crRNA (Supplementary Fig. 10), although there was a standalone product of ~26 nts representing the terminal repeat with ~10 nts trimmed from its 3′ end. This product is unlikely to bind to Cas12a given the lack of a formed stem loop structure recognized by the nuclease. The absence of the standalone product in *Francisella* (Supplementary Fig. 6C) also likely reflects differences in RNA stability between bacteria and mammalian cells.

We finally asked whether the extraneous crRNA could direct DNA cleavage by FnCas12a. To test this directly, we inserted the corresponding target sequence with a canonical PAM into a plasmid and measured clearance with the three-spacer FnCas12a arrays (PcF-1/3/2) shown to generate an extraneous crRNA (Fig. 6b). As expected, this array led to significant clearance of the plasmid with the expected target in comparison to the plasmid lacking the target ($p = 2 \times 10^{-5}$ for two-tailed $t$-test, $n = 4$) (Fig. 7b). Furthermore, plasmid clearance was negligible when the 3′ repeat of the three-spacer array was deleted or replaced with the native terminal repeat ($p = 0.36$ and $0.38$ for two-tailed $t$-test, $n = 4$). These results thus show that the extraneous crRNA can

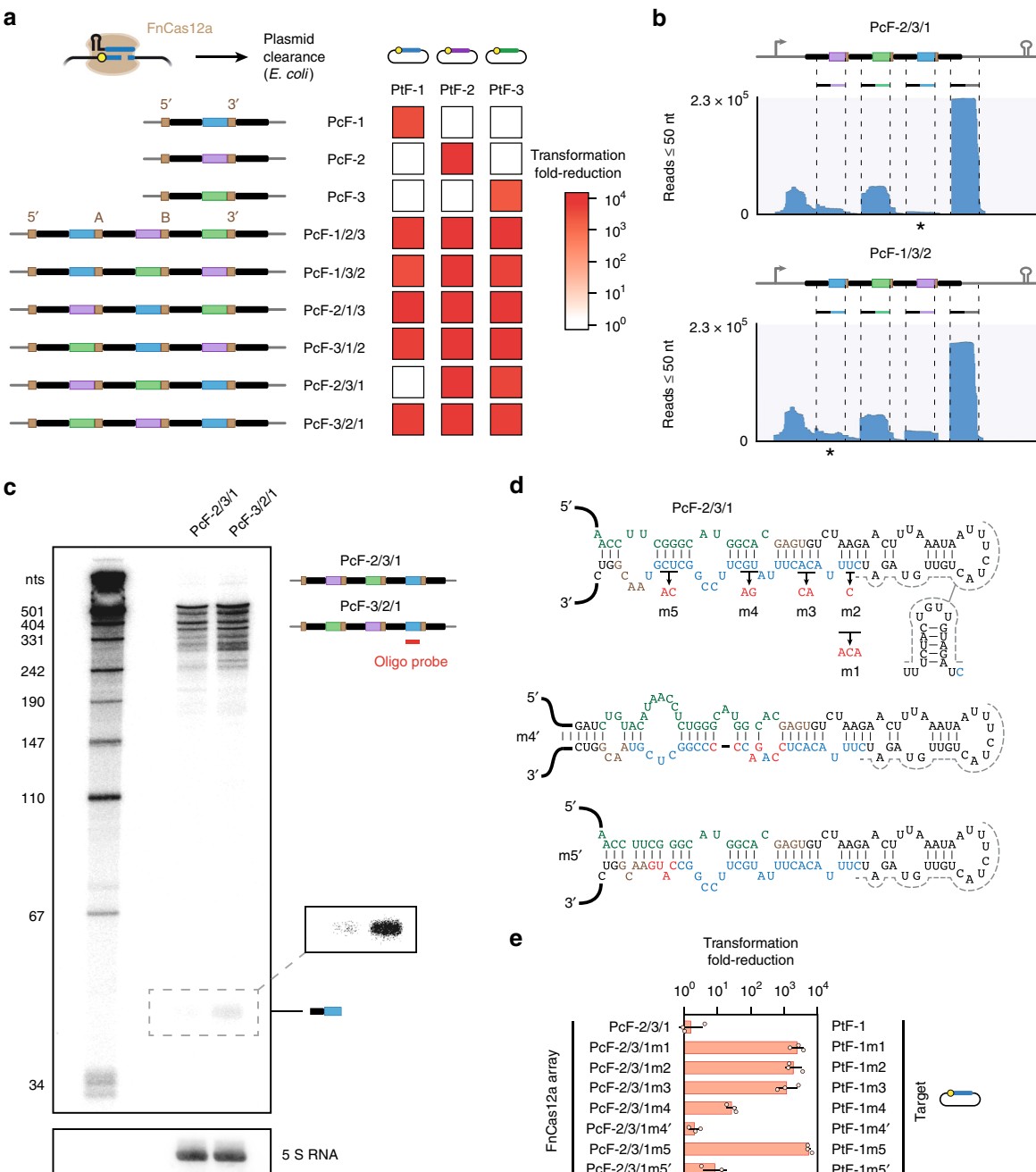

**Fig. 6** Loss of plasmid clearance linked to low crRNA abundance and global RNA secondary structure. **a** Multiplexed plasmid clearance by FnCas12a in *E. coli*. See Fig. 3a for details. Sequences of the spacers and junctions in the assembled arrays are shown in Supplementary Table 4. **b** RNA-seq analysis of the three-spacer arrays in PcF-2/3/1 and PcF-1/3/2. Each array and FnCas12a were co-expressed in TXTL prior to small-RNA isolation and RNA-seq analysis. See Fig. 5 for details. Stars indicate mapped reads for spacer S1. **c** Northern blotting analysis of the transcribed arrays from PcF-2/3/1 and PcF-3/2/1. Each array and FnCas12a were co-expressed in *E. coli* prior to total RNA isolation and Northern blotting analysis. The region in which the processed crRNAs should appear is shown with greater intensity. The analysis was conducted with a radiolabeled oligo probe complementary to spacer S1. Detection of 5S RNA was used as a loading control. Gel images from an independent experiment are in Supplementary Fig. 7. **d** Mutations made within a predicted imperfect hairpin between spacers S1 and S3. The predicted hairpin was present as a stable structure (base-pairing probabilities > 80%) only in PcF-2/3/1. Note that the structure prevents formation of the hairpin recognized by FnCas12a (gray dashed line). One set of mutations (m1– m5) were selected to disrupt the predicted secondary structure. Another set (m4′, m5′) represents expanded mutations to m4 and m5 that reform a stable secondary structure. Predictions of the minimal-free energy structure and base-pairing probabilities were made for the sequence spanning the 5′ repeat through the 3′ spacer. Text is colored to match that of spacer S3 (green), spacer S1 (blue), the assembly junctions (brown), and the intervening repeat (black). Mutations are in red. **e** Plasmid clearance directed by FnCas12a and the mutated versions of PcF-2/3/1. The protospacer in the target plasmid was mutated to match each changed spacer sequence. See Fig. 3a for details. Sequences of the spacers and junctions in the assembled arrays are shown in Supplementary Table 4. Values represent the geometric mean and S.D. from three independent transformations starting from separate colonies. Source data are provided as a Source Data file

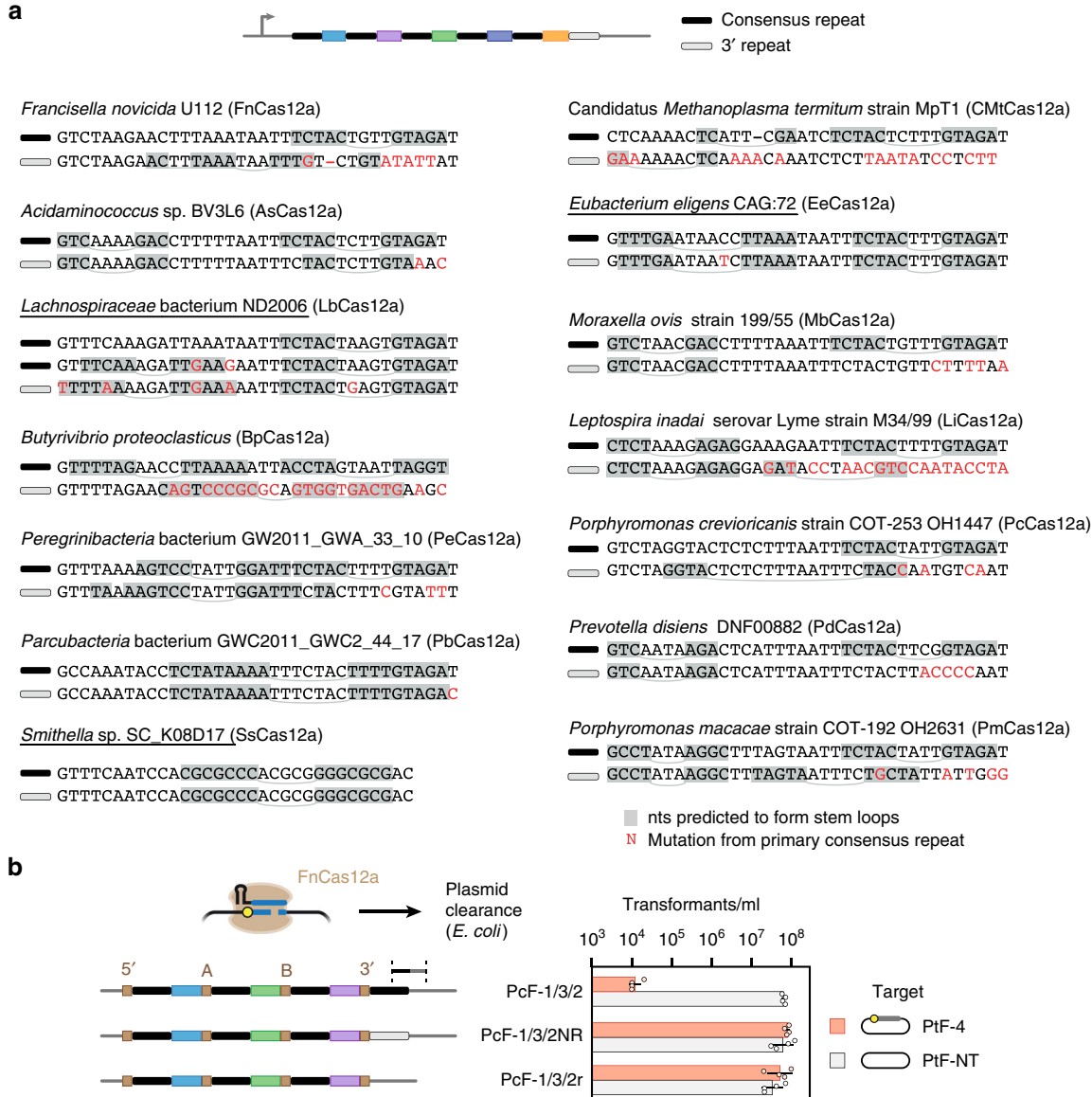

**Fig. 7** The native terminal repeats in Cas12a arrays prevent DNA targeting by the extraneous crRNA. **a** CRISPR-Cas12a terminal repeats often contain mutations expected to disrupt crRNA processing. CRISPR arrays from Type V-A CRISPR systems in 14 different bacteria were evaluated. For each CRISPR array, the terminal repeat is compared to the consensus repeat. In one example (*Lachnospiraceae* bacterium ND2006), two consensus sequences were present. Sequence deviations are highlighted in red, while base-paired stems predicted by NUPACK (http://www.nupack.org/) to form in the transcribed repeat are highlighted in gray. The thin gray lines show the loop (curved) or bulge (straight) of each predicted hairpin. Prior biochemical characterization of the repeat sequence revealed that the sequence and structure of the 3′ hairpin and the immediately upstream and downstream nts were critical for proper recognition and processing by Cas12a (refs. [15,58]). Underlined strains harbor terminal repeats that are not obviously mutated to disrupt processing. **b** Assessing plasmid clearance through the extraneous crRNA in *E. coli*. The clearance assays were conducted as described in Fig. 3a using PcF-1/2/3 with a consensus 3′ repeat, a native terminal repeat, or no 3′ repeat. Values represent the geometric mean and S.D. of three independent transformation experiments starting from separate colonies. Source data are provided as a Source Data file

actively direct targeting, and the native terminal repeat eliminates this activity.

## Discussion

Here, we devised and validated a technique called CRATES for the efficient, one-pot generation of CRISPR arrays and array libraries. The technique relies on assembling modular repeat-spacer subunits using defined junctions within the trimmed portion of the CRISPR spacers. By specifying the sequence and 5′ or 3′ directionality of each overhang of the junctions, we could minimize pairing between non-adjacent repeat-spacer subunits,

allowing the scalable production of larger CRISPR arrays. Furthermore, because the junctions are fully decoupled from the repeat sequence as well as the targeting portion of the spacer sequence, we could readily generate CRISPR arrays used by three different single-effector nucleases as well as a library of CRISPR arrays. The generated arrays are distinct from sgRNA arrays, as CRISPR arrays are much more compact and are not limited to Cas9.

Given the numerous applications of CRISPR technologies that could benefit from multiplexing with CRISPR arrays, CRATES has the potential to be widely implemented. One poignant example is using libraries of CRISPR arrays for combinatorial

genetic screens[49]. While CRISPR-based screens principally relied on a library of single sgRNAs, there are recent examples where libraries containing sgRNA pairs were used in human cells to identify synthetic-lethal genes as well as gene pairs that drive cancer proliferation or interact with tumor protein p53 (refs. [10,44]). In both examples, the sgRNAs had to be cloned sequentially. CRATES therefore could build on these approaches through the efficient, one-pot assembly of array libraries that extend beyond two targets, allowing the combinatorial screening of pathways with genetic redundancy. The libary we generated as a proof-of-principle demonstration exhibited a reasonable distribution uniformity given that the relative abundance of the majority of arrays (~80%) were within a 3-fold range and all of the arrays were within a seven-fold range. However, futher optimization may be needed when scaling to larger libraries.

CRATES also helped grant unique insights into factors influencing crRNA biogenesis and crRNA-directed nuclease activity. One insight was that FnCas12a could derive a fully functional crRNA from the terminal repeat in a CRISPR array. This observation was unexpected given that only spacers flanked by two repeats are normally expected to yield crRNAs. The more practical ramification is that the terminal repeat-derived crRNA could lead to unintended targeting. We provided evidence that terminal repeats for Cas12a arrays have been under negative selection to prevent formation of a crRNA, resulting in mutations within the terminal repeat that disrupt processing. A separate option is expressing Cas12a arrays without a terminal repeat (Fig. 7b)[15,50]. However, the 3′ repeat was shown to be important when deriving Cas12a crRNAs from eukaryotic mRNAs[14], and we recently showed that processing of the transcribed FnCas12a array was sensitive to adjacent stable secondary structures that was exacerbated in the absence of a 3′ repeat[48]. As this inhibitory effect was eliminated with a native terminal repeat, these mutated repeats could be readily used with synthetic Cas12a arrays as a simple solution to prevent extraneous crRNAs and adjacent, inhibitory structures. The more practical ramification is that the terminal repeat-derived crRNA could lead to unintended targeting, at least in *E. coli*. Future work should evaluate the activity of these crRNAs in other cellular contexts and whether a native terminal repeat or no terminal repeat better promotes multiplexed targeting by an array. We also note the accumulation of the trimmed repeat in HEK293T cells (Supplementary Fig. 10), indicating more work is needed to understand crRNA biogenesis in eukaryotic cells.

The second insight was the varying abundance of processed crRNAs and its connection to crRNA-directed nuclease activity and the predicted secondary structure of the transcribed array. While prior studies of natural CRISPR arrays reported varying crRNA abundance[7,15,33,46], it remained unclear why the abundance varied and whether the crRNAs exhibited compromised nuclease activity. The loss of nuclease activity that we observed was striking given that it was only observed for one of the six possible configurations of a three-spacer array, suggesting a dependence on the spacer's sequence and context. We also provided multiple lines of evidence for the role of global RNA secondary structure and misfolding of the hairpin recognized by Cas12a as part of crRNA biogenesis. Other factors such as cryptic promoters or RNase sites could be involved, and the influence of secondary structure is likely more complicated given coupling between transcription and crRNA processing and the folding of processing intermediates. However, the predicted secondary structure of the array was an indicator of crRNA abundance, at least in the few examples we tested. Finally, it remains to be seen how frequently a crRNA loses the ability to direct nuclease activity in the context of an array; prior work did not observe any effect on spacer order[16], although the number of tested arrays and

spacer sequences has been limited. Future work thus could extend these insights to develop design tools for synthetic CRISPR arrays exhibiting uniform crRNA abundances.

## Methods

**Strains and growth conditions**. Supplementary Table 2 provides a list of the key resources used in this work. All strains, plasmids, and oligonucleotides are listed in Supplementary Tables 3 and 4. For experiments in *E. coli* and TXTL, SpCas9 was expressed from pCas9 (Addgene #42876) while the other *cas* genes were expressed under the control of the J23108 promoter in a pBAD33 backbone. The targeted plasmids used with the plasmid clearance assays for FnCas12a were constructed by Q5 mutagenesis (New England Biolabs) following the manufacturer's instructions to remove portions of the pUA66-lacZ plasmid (GFP gene driven by *lacZ* promoter) so that each targeted plasmid only contain one protospacer matching to the designed spacers. The targeted plasmids used with the plasmid clearance assays for SpCas9 were constructed by inserting the fragments containing the PAM and protospacer in between the *XhoI* and *BamHI* sites of pUA66. The GFP reporter plasmids used with the gene repression assays were constructed by inserting a constitutive (*lacIQ*) or inducible promoter (*lacZ* or *araB*) into the *XhoI* and *BamHI* restriction sites upstream of the *gfp* gene in pUA66 (ref. [51]). The reporter plasmid for in vitro RNA detection with LsCas13a was the p70a-deGFP plasmid reported previously[52]. The target encoding plasmids were constructed by inserting the protospacer into the plasmid pUA66_PJ23119 by replacing the ORF of GFP gene. Plasmid p70a-deGFP was also used for the off-target binding experiments by inserting each target sequence to upstream of the P70a promoter using Q5 mutagenesis. The Cas12a encoding plasmids were constructed by inserting the open reading frame of FnCas12a or AsCas12a into pBAD33 with a constitutive promoter (PJ23108). The plasmids encoding dFnCas12a variants were constructed by Q5 mutagenesis using the FnCas12a plasmid as template. Plasmid used for expressing Cas9 was pCas9 (Addgene # 42876). Plasmid pCas9-notracr was constructed by removing the tracrRNA portion from the plasmid pCas9 using Q5 mutagenesis. The plasmid pcDNA3.1-hFnCpf1 (Addgene # 69976) was used to express FnCas12a in HEK293T cells. The in vivo plasmid clearance assays were conducted in CB414, a derivative of *E. coli* BW25113 with the *lacI* promoter through the *lacZ* gene and the endogenous I-E CRISPR-Cas system deleted.

*E. coli* cells were grown in Luria Bertani (LB) medium (10 g/L NaCl, 5 g/L yeast extract, 10 g/L tryptone) at 37 °C with shaking at 250 rpm. The antibiotics ampicillin, chloramphenicol, and/or kanamycin were added to maintain any plasmids at 50 μg/mL, 34 μg/mL, and 50 μg/mL, respectively. The inducers Isopropyl β-D-1-thiogalactopyranoside (IPTG) or L-arabinose were added at concentrations of 0.2 mM and 0.2% when specified.

The *S. cerevisiae* strain YPH500 (Stratagene) was used as the background strain for gene activation. Culturing and genetic transformation were done using standard laboratory techniques[53] using either URA3, HIS3, or LEU2 as selectable markers. The reporter plasmid for gene activation was constructed from integrative plasmid pRS406 (Stratagene) similar to prior work using standard techniques[54]. Specifically, a CEN pRS414 plasmid was used to express the FnCas12a variants from a GPD promoter. To construct FnCas12a-VP64, the FnCas12a double mutant was cloned in between pGPD and VP64 in pRS414 through Gibson assembly. All plasmid constructs were generated using TOP10, Novablue, Tg1, or DH5α electrocompetent cells and verified by PCR and Sanger Sequencing of the inserted sequence.

**One-pot generation of CRISPR arrays**. The backbone plasmid used for generating CRISPR arrays for FnCas12a (pFnCpf1GG) was constructed by Gibson assembly to join three PCR fragments together and kill the extra BsmBI site on the scaffold backbone. The three PCR fragments contain a constitutive J23119 promoter, a GFP-dropout construct (with promoter and terminator) flanked by two Type IIS BsmBI restriction sites and a direct repeat of FnCas12a, and *rrnB* terminator, ampicillin resistance gene, and pMB1 origin of replication, respectively. The backbone plasmid used for generating arrays for AsCas12a, and LsCas13a, and the four-spacer composite arrays were constructed using Q5 mutagenesis to remove the FnCas12a repeat and insert the other corresponding repeats. The backbone plasmid used for generating arrays for SpCas9 was constructed by inserting a PCR fragment with the Cas9 direct repeat and BsmBI sites flanked mRFP dropout construct into AatII and HindIII digested pFnCpf1GG. Backbone plasmid for generating composite arrays for blocking off-target cleavage was constructed using Q5 mutagenesis to remove extra nucleotides on the 3′ of the promoter from pFnCpf1GG, so that the transcription starts from the sgRNA of Cas9. Backbone plasmid for generating arrays to use in yeast was constructed by using Gibson Assembly to join three PCR fragments and mutate a BsaI site on the scaffold backbone: BsaI sites flanked GFP-dropout construct and two PCR fragments amplified from plasmid pRS413, on which arrays are transcribed from a SNR52 snoRNA promoter. The sequences of the resulting backbone plasmids are available in Supplementary Table 2. Forward and reverse oligonucleotides encoding one repeat, one spacer, and a 4-nt junction were phosphorylated and annealed to form dsDNA with a 5′ and/or 3′ overhangs. In all, 400 fmol of each dsDNA, 20 fmol of backbone plasmid, 1 μL of T4 ligase, and 1 μL of BsmBI or BsaI were added to 2 μL of T4 ligation buffer, then water was added to reach a total volume of 20 μL.

A thermocycler was used to perform 25 cycles of digestion and ligation (42 °C for 2 min, 16 °C for 5 min) followed by a final digestion step (60 °C for 10 min) and a heat inactivation step (80 °C for 10 min). The ligation mix was then diluted 1:6 in water and electroporated into competent *E. coli* cells. After transformation and recovery for 1 h at 37 °C with shaking at 250 rpm in SOC media, cells were plated on LB agar containing the appropriate antibiotic and incubated for 16 h. White colonies were then screened for the presence of the correct band size, and the array was validated through Sanger sequencing of the PCR product. See Supplementary Methods for a detailed protocol and an example of assembling one of the three-spacer FnCas12a arrays.

The no-spacer control used in most of the experiments was generated by inserting a single repeat into the backbone plasmid, resulting in two consecutive repeats with no intervening spacer.

Composite arrays for coordinated plasmid clearance and gene repression were generated using the same method except that the backbone plasmid had a 5' direct repeat for SpCas9 and a 3' direct repeat for FnCas12a. Composite arrays for reducing off-targeting were generated by assembling the Cas9 sgRNA and two repeat-spacer dsDNAs into the GFP drop-out backbone. The sgRNA was formed with two annealed oligonucleotides, in line with CRATES.

Plasmids encoding seven-spacer arrays driven by CMV promoters were constructed by Gibson assembly using three DNA fragments each. One fragment was amplified from plasmid pCB871, which contained the seven-spacer array generated using CRATES with consensus repeat as terminal repeat or with a native terminal repeat. The other two fragments were the backbone amplified from plasmid pMZ-DF-JEV, which is a gift from Dr. Neva Caliskan.

**One-pot generation of the CRISPR array library**. pFnCpf1GG was used as the backbone to generate the CRISPR array library. To make a library of three-spacer arrays with five spacer variants in each spacer locus, thirty forward and reverse oligonucleotides each encoding one repeat, one spacer, and a 4-nt junction were synthesized from Integrated Device Technology (IDT). As part of the library design, the 5' and 3' assembly junction sequences for all repeat-spacer pairs destined for the same position in the array (e.g., spacers S1a-e destined for the 5' position in the three-spacer array included annealed oligos with a 5' CCCT overhang at the 5' end and a 3' GCTG overhang at the 3' end). Otherwise, the oligo design followed exactly that specified under "One-pot generation of CRISPR arrays." Supplementary Table 4 contains the specific oligonucleotides and assembly junctions used for the library generation. The oligos were phosphorylated and annealed to form dsDNA with a 5' and/or 3' overhangs. In all, 400 fmol of each dsDNA, 20 fmol of backbone plasmid, 0.5 µL of T4 ligase (1000 units), and 1.5 µL of BsmBI (15 units) were added to 2 µL of 10x T4 ligation buffer, then water was added to a total volume of 20 µL. A thermocycler was used to perform 35 cycles of digestion and ligation (42 °C for 2 min, 16 °C for 5 min) followed by a final digestion step (60 °C for 10 min) and a heat inactivation step (80 °C for 10 min). Five microliters of the ligation mix was then transformed into chemical competent *E. coli* cells. Note that purified and concentrated ligation product and electrocompetent *E. coli* cell could be used to improve transformation efficiency for generating large library. After transformation and recovery in 500 µL SOC for 1 h at 37 °C with shaking at 250 rpm, 10 µL cells were plated on LB agar containing the appropriate antibiotic and incubated for 16 h to check the number and color of the resulting colonies. The rest of the recovered culture was added to 15 mL LB media containing the appropriate antibiotic and incubated at 37 °C with shaking at 250 rpm to OD$_{600}$ ≈ 1. Cells were harvested by centrifugation and subjected to plasmid extraction. The extracted plasmids were used as a template for PCR to generate the DNA library for next-generation sequencing.

**Plasmid clearance in *E. coli***. We transformed 50 ng of the plasmid encoding the CRISPR array or a non-targeting control with no spacer into *E. coli* cells harboring a plasmid encoding a Cas protein and another plasmid encoding the crRNA target sequence. After recovering for one hour in SOC at 37 °C with shaking at 250 rpm, cells were serially diluted and 10 µL of droplet was plated on LB agar plates with ampicillin, kanamycin, and chloramphenicol. After 16 h of growth, colony numbers were recorded for analysis.

**Gene repression in *E. coli***. CB414 cells were initially transformed with three compatible plasmids: a plasmid encoding a variant of FnCas12a, a plasmid encoding a CRISPR array or no-spacer control, and a plasmid encoding GFP under the control of the *lacZ*, *lacIQ*, or *araB* promoter. Overnight cultures of cells harboring the three plasmids were back-diluted to ABS$_{600}$ ~0.01 in LB medium with ampicillin, kanamycin and chloramphenicol and the promoter's inducer and shaken at 250 rpm at 37 °C until ABS$_{600}$ reach ~0.2. Cultures were then diluted 1:25 in 1x phosphate-buffered saline (PBS) and analyzed on an Accuri C6 flow cytometer with C6 sampler plate loader (Becton Dickinson) equipped with CFlow plate sampler, a 488-nm laser, and a 530+/− 15-nm bandpass filter. GFP fluorescence was measured similar to prior work[55]. Briefly, forward scatter (cutoff of 15,500) and side scatter (cutoff of 600) were used to eliminate non-cellular events. The mean value within FL1-H of 30,000 events within a gate set for *E. coli* were used for data analysis[56].

**In vitro multiplexed RNA detection with Cas13a**. Open reading frame of LsCas13a was cloned into pBAD33 with a constitutive promoter. Single or multiple spacer arrays were generated as described above. Targeted RNAs were transcribed using a constitutive promoter on the plasmid. A plasmid constitutively express a deGFP protein was used as reporter. The Cas13a encoding plasmid, array encoding plasmid, and target encoding plasmid were added into 9 µL of *MyTXTL* master mix (Arbor Biosciences) to a final concentration of 2 nM, 1 nM, and 0.5 nM, respectively, to a total volume of 12 µL, and incubated at 37 °C for 2 h. Then the reporter plasmid was added to a final concentration of 0.5 nM. Aliquots of 5 µL were placed into the wells of 96-well V-bottom plate (Corning Costar 3357) and incubated at 37 °C for 16 h in a Synergy H1MF microplate reader (BioTek) with kinetic reading every 3 min (excitation, emission: 485 nm, 528 nm; gain: 60; lightsource: Xenon Flash).

**Gene activation in *Saccharomyces cerevisiae***. Three single yeast colonies for each strain were picked after plasmid transformations and inoculated into 500 mL of SD-media (synthetic dropout media containing 2% glucose with defined amino acid mixtures) in Costar 96-well assay blocks (V-bottom; 2 mL max volume; Fisher Scientific). The cultures were grown at 30 °C with 250 rpm shaking for 24–48 h. Cultures were then re-inoculated in SD-complete media to an OD$_{600}$ of 0.05 to 0.1 and grown at 30 °C with 250 rpm shaking for 12 h. Cells were treated with 10 mg/mL cycloheximide to inhibit protein synthesis and then assayed for yEGFP expression by flow cytometry. A total of 10,000 events were acquired using a MACSquant VYB flow cytometer with 96-well plate sampler. Events were gated by forward scatter and side scatter and all values obtained were from three isogenic strains. Plots were generated based on side scatter versus FL1 fluorescence. See Supplementary Fig. 3D for representative plots.

**In vitro assessment of blocking off-target cleavage**. Single guides for Cas9 cleavage, single-spacer arrays for dFnCpf1 blocking off-target sites, and composite arrays for cleavage and blocking synchronously were generated as described under "One-pot generation of CRISPR arrays". The protospacers were cloned into upstream of constitutive promoter of GFP gene on a plasmid. Plasmid encoding dFnCpf1, targeting plasmid for Cas9, targeting plasmid for dFnCpf1, composite array Psg/b1/b2, composite array Psg/b2/b1 or non-targeting control, and one of the three targeted plasmids were added into *MyTXTL* master mix to final concentration of 1 nM, 2 nM, and 0.5 nM, respectively, and incubated at 29 °C for 4 h. Then the SpCas9 encoding plasmid (pCas9-notracr) was added to the mixture to a final concentration of 2 nM and incubated at 29 °C for 16 h in a microplate reader with kinetic reading (excitation, emission: 485 nm, 528 nm) every 3 min. Non-targeting control was also tested as a control.

The off-target blocking was also assessed in TXTL when the on- and off-target sites existed in one pot using next-generation sequencing. Plasmid encoding dFnCpf1, composite array Psg/b1/b2, composite array psg/b2/b1 or non-targeting control, and all of the three targeted plasmids were added into *MyTXTL* master mix to final concentration of 1 nM, 2 nM, and 0.5 nM, respectively, and incubated at 29 °C for 4 h. Then the SpCas9 encoding plasmid (pCas9-notracr) was added to the mixture to a final concentration of 2 nM and incubated at 29 °C for 8 h. Plasmids were extracted from the TXTL reaction and used as template for PCR to make DNA library for next-generation sequencing.

**crRNA generation in TXTL**. Plasmids encoding FnCas12a and the arrays were added into 9 µL of *MyTXTL* master mix (Arbor Biosciences) to a final concentration of 5 nM each in a PCR tube and a total volume of 12 µL. Two micro-molar of Chi6 annealed oligos was included in the reaction to prevent any unintentional degradation of the plasmids by RecBCD proteins. The mixture was incubated at 29 °C for five hours in a thermocycler, and total RNA was extracted using Direct-zol RNA MiniPrep kit following the manufacturer's instructions (Zymo Research).

**Mammalian cell culture and transfection**. Human embryonic kidney 293T (HEK293T) cells (a kind gift from the Redmond Smyth lab) were grown at 37 °C with 5% $CO_2$ in Dulbecco's modified Eagle's medium (DMEM, Gibco) containing 10% iron-supplemented calf serum (Sigma) and 1% penicillin–streptomycin (Gibco). Cells were seeded in six-well plates (Corning) and transfected at 50% confluency using 1 mg/mL Polyethylenimine (PEI, Polysciences) prepared following the manufacturer's instructions. Co-transfections were performed by adding to each well 100 µL of DMEM containing 300 ng FnCpf1 encoding plasmid, 300 ng CRISPR array encoding plasmid and 7.2 µL of 1 mg/mL PEI. Forty-eight hours after transfection, cells were washed with phosphate-buffered saline (PBS, Gibco) and directly lysed using 900 µL TRIzol (Zymo Research) per well. Total RNA was extracted using Direct-zol RNA MiniPrep kit (Zymo Research) following the manufacturer's instructions.

**Small-RNA library preparation sequencing and data analysis**. The extracted Total RNA was treated with Turbo DNase (Life Technologies), and 3' dephosphorylated with T4 Polynucleotide Kinase (New England Biolabs), rRNAs were then depleted using Ribo-Zero Kit (Bacteria or H/M/R) (Illumina) following the manufacturer's instructions. RNA libraries were prepared using NEBNext

Multiplex Small RNA Library Prep Set for Illumina (New England Biolabs) following the manufacturer's instructions. Samples were sequenced on a MiSeq machine (Illumina) by the Genomic Science Laboratory at North Carolina State University. The Geneious 10.2.3 software package (Biomatters) was used for data sorting and alignment. Briefly, Fastq reads were trimmed and quality filtered using the BBDuk plugin. Trimmed reads were then aligned to created CRISPR array reference sequences using Geneious 10.2.3′ Map to Reference.

**Next-generation sequencing and data analysis**. DNA regions of interest were amplified from extracted plasmids. Primers used contained the necessary adapters for analysis by Illumina MiSeq:

Fwd 5′-TCGTCGGCAGCGTCAGATGTGTATAAGAGACAG-3′
Rev 5′-GTCTCGTGGGCTCGGAGATGTGTATAAGAGACAG-3′

The amplicons were purified using Select-a-Size DNA Clean & Concentrator (Spin-Column) (Zymo Research) following the manufacturer's instructions to remove excess primers and possible primer dimers. The purified DNA was then indexed through eight cycles of amplification using Nextera indexes E502, E503, E504, E505, and N707. The resulting DNA products were again purified using Select-a-Size DNA Clean & Concentrator (Spin-Column) (Zymo Research) following the manufacturer's instructions. Samples with different indexes were pooled together as 10 nM in a total volume of 20 μL and submitted to Core Unit Systemmedizin (Würzburg, Germany) for paired-end 2 × 150 bp deep sequencing. Resulting sequencing data was sorted according to the indexes and trimmed for quantifying each arrays using Salmon on the platform Galaxy Version 0.9.1 (https://usegalaxy.org/). Salmon was developed as a tool for transcript quantification from RNA-seq data. we re-purposed it to quantify the abundance of different arrays by querying the DNA-reads against the reference sequences of 125 arrays we aimed to assemble. KmerLen was set as 31 and quasi was chosen as the type of index to build[57].

**Northern blotting analysis**. CB414 cells were initially transformed with two compatible plasmids: the FnCas12a plasmid and the array plasmid PcF-2/3/1 or PcF-3/2/1. Overnight cultures of cells harboring the two plasmids were back-diluted to an $ABS_{600}$ of ~0.01 in LB medium with ampicillin and chloramphenicol and shaken at 250 rpm at 37 °C until the $ABS_{600}$ reach ~0.6. Cells were harvested by centrifugation, and total RNA was extracted using Direct-zol reagent and the Direct-zol RNA MiniPrep kit (Zymo Research) following the manufacturer's instructions with on-column RNase treatment following the manufacturer's instructions. The extracted total RNA was further treated with Turbo DNase (Life Technologies). For Northern blotting analysis, 5 μg of each RNA sample was resolved on an 8% polyacrylamide gel containing 7 M urea at 300 V for 140 min using a gel transfer system (Doppel-Gelsystem Twin L, PerfectBlue). RNA was transferred onto Hybond-XL membranes (Amersham Hybond-XL, GE Healthcare) using an Electroblotter with an applied voltage of 50 V for 1 h at 4 °C (Tank-Elektroblotter Web M, PerfectBlue), crosslinked with UV-light for a total of 0.12 Joules (UV-lamp T8C; 254 nm, 8 W), hybridized overnight in 17 ml Roti-Hybri-Quick buffer at 42 °C with 5 μL γ-32P-ATP end-labeled oligodeoxyribonucleotides (Table S1) and visualized using a Phosphorimager (Typhoon FLA 7000, GE Healthcare). Gel images from a replicate experiment are shown in Fig. S7.

**Plasmid transformation in *E. coli***. Fold reduction was calculated as the ratio of colony-forming units (CFU's) for cells transformed with the no-spacer array plasmid over that for cells transformed with the CRISPR array plasmid.

**Fluorescence measurements in *E. coli***. GFP fold-repression was calculated using mean fluorescence values, subtracting the fluorescence of *E. coli* cells lacking the GFP reporter plasmid, and dividing the resulting fluorescence value for the no-spacer array by that for the tested CRISPR array.

**Fluorescence measurements in yeast**. The fraction of GFP-positive cells were calculated by setting a threshold on FL1 so < 0.1% of the cells lacking a crRNA fall within the GFP-positive bin. The reported fraction is the percentage of cells that fall above the threshold within the entire gated population.

**RNA sensing in TXTL**. The relative levels of GFP (%GFP) were calculated as the ratio of the fluorescence value divided by that using PtL-nt with the same targeting array construct.

**Blocking off-target cleavage in TXTL**. The transcriptional rate was calculated as the slope of each fluorescence signal curve from 2.5 h to 4.5 h after the pCas9-notracr plasmid was added. Protection percentage was calculated as the ratio of the transcriptional rate for CRISPR array over that of the non-targeting control.

**Reporting summary**. Further information on research design is available in the Nature Research Reporting Summary linked to this article.

## Data availability

Next-generation sequencing data is available through NCBI sequence read archive (SRA). The RNA-seq analysis of crRNA abundance is accessible through SRA accession numbers SRP144980 and PRJNA454865. Next-generation sequencing data for the CRISPR array library is accessible through SRA accession number PRJNA496034. See Supplementary Table 3 for the the list of all plasmids obtained from Addgene. All other data are available upon reasonable request.

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

## Acknowledgements

We thank Jörg Vogel and Lars Barquist for critical comments. We also thank Redmond Smyth for the use of his lab as part of the experiments with mammalian cells. The pCas9 plasmid was a gift from Luciano Marraffini (Addgene # 42876), the pcDNA3.1-hFnCpf1 plasmid was a gift from the Zhang lab (Addgene # 69976), and the plasmid pMZ-DF-JEV was a gift from the Caliskan lab. The work was supported through funding from the NIH (1R35GM119561 to C.L.B. and 1DP1DA044359 to A.J.K.), the North Carolina State University Summer Undergraduate Research Grant (to T.D.C.), Agilent Technologies (Gift #3926 to C.L.B.), and the Camille & Henry Dreyfus Foundation (2017-137 to C.L.B.).

## Author contributions

C.L. and C.L.B. conceived this study. C.L. and C.L.B. designed the experiments. C.L., F.T., R.A.S., and S.R.D. performed the array cloning and experiments with bacteria and TXTL; C.L. made the RNA library and DNA library for next-generation sequencing; C.L. and C.L.B. analyzed the data. A.J.K. and T.D.C. conducted the experiments with yeast. R.T.L. analyzed the RNA sequencing data. C.L. and C.L.B. wrote the manuscript, which was read and approved by all authors.

## Additional information

**Competing interests:** C.L.B. is a co-founder and scientific advisory board member of Locus Biosciences and has submitted provisional patent applications on CRISPR technologies. The other authors declare no competing interests.

