## [Peer Review File · Nature Communications]

Reviewers' Comments:

Reviewer #1:

Remarks to the Author:

In this study, Liao et al. describe a strategy for constructing CRISPR RNA (crRNA) arrays and array libraries, which they use to investigate various aspects of crRNA biogenesis and for multiplex targeting of Cas nucleases. The approach, termed CRATES for CRISPR assembly through trimmed ends of spacers, is a one-pot cloning scheme for crRNA arrays whereby assembly overhangs are positioned within the trimmed portion of each spacer. CRATES was validated for three Cas nucleases in bacteria and yeast, and used to generate a 125-variant library of three-spacer Cas12a arrays. The authors also generate 'composite' arrays containing two SpCas9 spacer subunits and two FnCas12a subunits, which were used in vitro to reduce off-target cleavage by directing the catalytically inactive FnCas12a to predicted off-target sites of SpCas9. RNA-seq analysis of small RNAs generated by in vitro transcription of a seven-spacer FnCas12a array led to the observation that the terminal repeat generates an unwanted crRNA-like short RNA. The authors suggest that this phenomenon is prevented in native bacterial CRISPR-Cas arrays by the accumulation of mutations in the terminal repeat. Finally, the authors describe a relationship between the secondary structure of transcribed CRATES arrays and the abundance of processed crRNAs. This could be extended to produce a set of design rules to prevent secondary-structure formation in spacer-subunit arrays.

This manuscript raises interesting questions about the biogenesis of crRNAs in native and engineered systems – these insights may prove useful in the design of crRNA arrays for multiplex CRISPR-based applications. However, on its own, CRATES is neither sufficiently innovative in terms of technology development, nor does it address a major technical limitation in the field. Furthermore, with regard to fundamental biological discoveries, the manuscript suffers from several conceptual and technical flaws (detailed below), which cast doubt on many of the insights described by the authors. In particular, most conclusions are based on circumstantial evidence or in vitro / in silico assays, and no experiments are carried out in mammalian cells. Therefore, the manuscript in its present form does not provide sufficient novelty or biological insight to warrant publication in Nature Communications.

Conceptual issues

1. The paper suffers from a framing issue: the authors place excessive emphasis on CRATES as a novel approach for constructing spacer-subunit arrays. In fact, Gibson assembly and Golden gate cloning are routine techniques that can be used for the assembly of crRNA arrays, and direct synthesis is an effective (and relatively affordable) method for generation of scarless crRNA array libraries. Therefore, all experiments presented in Figures 3-6 would have been possible without CRATES. CRATES is simply a cloning method and as such should not be the main focus of the manuscript – the technique can be described in 1-2 paragraphs in the main text, and more details can be provided in the Methods section. The focus of the paper would seem better placed on the study of crRNA biogenesis and the possible applications of crRNA arrays. In addition, it should also be noted that although CRATES provides a low-cost alternative for assembling array libraries, it does appear to suffer from a very broad distribution of crRNA variant abundance.
2. Although the implementation of engineered CRISPR systems in bacteria and yeast is of interest, the majority of current research involving Cas and dCas programmable proteins is focused on engineering mammalian and plant cells. It is not clear why the authors have only tested CRATES arrays in bacteria and yeast where the practical value is reduced compared to other systems. Although certain dimensions of this study, such as processing of Cas9 crRNA arrays, would be significantly more difficult in eukaryotic cells (since unlike Cas12a and Cas13, Cas9 cannot process its own crRNA arrays), CRATES-generated CRISPR arrays should be tested and the proposed mechanistic insights should be validated in mammalian cells.
3. Figure S3A describes Cas12-inactivating mutations but this analysis does not seem relevant to

the theme of the paper. Furthermore, the behaviour of these mutants does not seem to correlate with previous studies. As stated in the manuscript, the D1255A mutant should only partially disrupt catalytic activity – instead, all mutants appear to have the same deleterious effect. In addition, E1006A shows a significant loss of DNA binding and, surprisingly, additional mutations appear to restore its activity. This effect is very difficult to reconcile and although the authors acknowledge it, they do not offer any explanation for this surprising observation. This leaves the reader to believe that either the story is incomplete or the TXTL assay is unreliable.

4. The authors observe unintended crRNA production from terminal repeats in FnCas12a arrays, and suggest that this could titrate the nuclease and lead to off-target DNA damage – however, this phenomenon is only described *in vitro* using the TXTL assay. They infer from sequence analysis that the same would occur in bacteria. However, it is unclear whether this occurs in mammalian cells and, if so, whether errant crRNAs would actually load into Cas12a and lead to mistargeted nuclease activity. Furthermore, the authors should test their assertion that naturally-occurring terminal repeats are mutated to prevent this phenomenon. For example, they could introduce one of these mutations in the last repeat of the synthetic FnCas12a array and assess crRNA generation by RNA-seq. Finally, since removing the terminal repeat does not seem to interfere with the activity of the PcF-1/3/2 array (Figure S6), it suggests that this issue could be simply avoided by creating arrays lacking the last repeat sequence.

Technical concerns

1. Although the authors describe their quantification analysis, it is impossible to infer error or significance from heat maps. The heatmaps should be accompanied by conventional plots containing error bars and appropriate statistical tests.
2. The Sanger sequencing data supporting the claim that assembly scars are trimmed during processing is not provided (this result is hidden in NGS data in Figure 5). Since this assumption lies at the foundation of CRATES, this data should be provided in the supplementary figures.
3. Sequentially delivering dFnCas12a and SpCas9 to a cell-free TXTL system is not an appropriate method for assessing nuclease protection in a cellular environment (Figures 4D, S4A-C). These experiments should be performed in cells (ideally in mammalian cells) using a competition assay whereby different fluorescent reporters are coupled to the on-target and off-target sites, and the activity of SpCas9 is assessed in parallel at each site in the presence or absence of dFnCas12a.
4. The analysis of crRNA secondary structure relies entirely on output generated by the predictive algorithm NUPACK. Since RNA secondary structure varies between species and cell types due to the cellular and protein environment, *in silico* modelling is only predictive of a certain fold. Therefore, it is difficult to extrapolate results from *in silico* predictions into bacteria or mammalian cells. In addition, the analysis is further weakened by the lack of proper randomised controls to establish the baseline folding capacity of similar length RNA sequences. For example, the authors should randomly shuffle the array sequence (thousands of iterations) and predict folding *in silico* to determine the background mean and standard deviation contribution to secondary structure at each nucleotide position. Finally, these predictions should be tested in cells by constructing different synthetic arrays and comparing their activity (as in Figure 6E).
5. The claim regarding context-dependent loss of crRNA-directed FnCas12a activity is only supported by the analysis of two arrays – the single array that displayed poor plasmid clearance activity (PcF-2/3/1), and one that was very effective (PcF-1/3/2). To establish whether the imperfect global secondary structure of the array (i.e. disruption of the repeat hairpin) is responsible for this phenomenon, these analyses should be carried out using the other four array combinations that were effective at plasmid clearance. To substantiate this conclusion, it is crucial to determine spacer production from the PcF 1/2/3, 2/1/3, 3/1/2, and 3/2/1 arrays, and compare the secondary structure of various hairpins across the arrays. In addition, since the band intensities in the Northern blot are very weak, replicate blots should be provided as supplemental

figures and band intensities should be quantified for statistical analysis.

Minor issues

1. Panel 2D is not referenced in the text – it should be referenced on line 128.
2. Line 140 should read “AsCas12a”.
3. Missing reference in line 182.
4. Line 203 states that the dFnCas12a double mutant was used for the VP64 fusion – in Figure 3C and the associated legend it is stated that the triple mutant was used. The authors should clarify this.
5. Figure 3C –spacers PcL-nt and PcL-1 clearly display non-specific activity (in particular at PtL3 target) – this is ignored in the text. The authors should explain this phenomenon and provide the raw data and associated quantification.
6. Flow cytometry gating strategies for yeast experiments should be included as supplemental data.
7. The variant library cloning scheme is not explained in sufficient detail.
8. A positive readout for measuring the efficiency of CRATES arrays (such as flow cytometry using a fluorescence reporter) would be more reliable than a negative readout (i.e. loss of antibiotic resistance).
9. The authors should clarify that the experiments in Figure 3B and 4 rely on the host bacterial nucleases to process the Cas9 spacer-subunit array.
10. Figure legend S4 should read “see Figure 4E” instead of “see Figure 5E”.
11. In Figure 6B, the authors compare arrays PcF-2/3/1 and PcF-1/3/2; yet, in Figure 6C, they switch the effective array to PcF-3/2/1, without providing any explanation or mentioning this in the text. As discussed above, all these analyses (RNA-seq, Northern blot and RNA secondary structure prediction) should be carried out for all array combinations.

Reviewer #2:

Remarks to the Author:

Liao and co-workers describe the development of a one-pot assembly approach for CRISPR arrays. This straightforward method appears to simplify the generation of design arrays for multiplexing by CRISPR associated nucleases, or even sets of these enzymes. Apart from the CRISPR assembly, this study describes a systematic investigation of the functionality of CRISPR-encoded guides, revealing the influence of context in the precursor-crRNA, most likely reflecting an important role of secondary structure. This paper seems well executed, and very well described and illustrated. The reported information will be of interest for the broad genome editing community. There are some issues that need attention.

Comments

1. In lines 27- 29 it is mentioned “joining sgRNAs with intervening cleavage domains to form sgRNA arrays”. It would be appropriate to cite Ferreira et al. ACS Synth Biol (2018).
2. Line 31-33: “yet the requirement for a tracrRNA and RNase III limited the use of CRISPR arrays utilized by Cas9 to bacterial applications”. This sentence should be adjusted. There are many examples in the literature of eukaryotic organisms which make use of the CRISPR arrays utilized by Cas9 with tracrRNA and endogenous RNase III. See Cong et al. Science (2013), as well as Mali et al. Science (2013) . Also, it should be mentioned that a synthetic single guide does not require RNaseIII (Jinek et al. Science (2012)).
3. Line 34: should read “process precursor-crRNAs”
4. Line 36: would be appropriate to add Reference of 4-gene multiplexing in yeast: Swiat et al. NAR (2017)).
5. Line 44: typo - 'for even for' > 'even for'
6. Figure 1B: not all repeat-spacer subunits are designed in the same way, meaning that some have 3'and 5'overhangs while others only 5'overhangs. This does not match with what is stated in

lines 90 and 91: "Repeat-spacer subunits were designed to contain alternating 5' and 3' overhangs"

7. Line 110: it is mentioned that arrays containing 7 spacers were correct 60% of the times. Did you sequence the false positive colonies - are they spontaneous deletions of *gfp*? Does the efficiency drop because of the chosen 4 bp linkers or because of the length of the array? Please comment.

8. References 13 and 31 are the same.

9. Line 139-142: Clarify if this is a different experiment from the one described in lines 142-148. Was this first experiment done with FnCas12a or AsCas12a?

10. Lines 181-182: "Cas12a nucleases can process transcribed CRISPR arrays in eukaryotic cells without accessory factors". This feature is not exclusive for eukaryotic cells; please rephrase.

11. Line 187: since this experiment is carried out in yeast, it should be mentioned that gene regulation has also been reported using the dead version of LbCas12a in yeast combined with different activators (VP64 (V), VP64-p65AD (VP), and VP64-p65AD-Rta (VPR)) (Lian et al. Nat Commun (2017)). This is also applicable for justifying lines 202- 204.

12. Lines 243-245: State the importance of this finding and compare it with other research. The swapping was only done for Cas9 spacers. Worth to include an extra experiment in which Cas12a spacers are swapped. In the experiment where the blocking of off-target sites is done, you swap the order but in this way, you could explore whether the context of the spacers can impact the resulting activity and if this phenomenon is sequence specific.

13. Lines 319-328: "CRATES revealed context-dependent loss of crRNA-directed FnCas12a activity". It is context dependent but also sequence dependent. For this reason, this result does not match with previous results that state that the context of the spacer in the array does not affect its activity. Please rephrase.

14. All the results state: "CRATES ...". CRATES is the assembling method. All the results that are obtained do not have to do with the assembling method itself but are observations of the activity of the arrays... . Please rephrase.

15. Discussion: it is important that it is stated that the results are spacer-sequence dependent (as indicated in the end of the discussion when talking about future design tools) . Please rephrase.

16. Line 688 of methods: "in SOC media" is repeated. Please rephrase.

We thank the reviewers for their thorough review of our manuscript and for their thoughtful feedback. We also thank the editor for the opportunity to submit a revised version. We provide point-by-point responses below in bold. Accompanying changes in the main text are in red.

As part of our revisions we completed the following major experiments that have been added to the manuscript. A summary of the experiments and their results are below:

- **Experiment #1:** We evaluated crRNA biogenesis by FnCas12a from the seven-spacer array in mammalian cells using RNA-seq analysis. These experiments garnered multiple insights that added to the overall story and our conclusions: (1) FnCas12a can process a CMV-driven CRISPR array assembled through CRATES into individual gRNAs, (2) the abundance of the resulting gRNAs paralleled that observed in TXTL, (3) a full-length unintended gRNA is generated in mammalian cells with a 3' consensus repeat but not a native terminal repeat. These results show that the insights concerning gRNA biogenesis extend to mammalian cells. Location of results: Figures 5 and S10.
- **Experiment #2:** We evaluated the targeting activity of the unintended gRNAs derived from the 3' end of the FnCas12a array. Using the plasmid clearance assay in *E. coli*, we found that the unintended crRNA led to plasmid clearance in the presence but not the absence of the expected target sequence. Furthermore, clearance was lost when the 3' repeat was deleted or replaced with the native terminal repeat. These results show that the 3' repeat can give rise to an active gRNA that is compromised with a native terminal repeat. Location of results: Figure 7.
- **Experiment #3:** We used NUPACK to predict the global RNA secondary structure of all six three-spacer arrays in the plasmid clearance assays with FnCas12a. The array lacking targeting activity with spacer S1 (PcF-2/3/1) was the only array to exhibit a predicted minimal-free-energy structure in which a repeat (the repeat associated with spacer S1) was misfolded with high base-pairing probabilities (>0.8). These structures provide further support for the loss of targeting activity because of misfolding of the transcribed CRISPR array and sufficiently motivated the mutational analyses that further confirmed the role of structure formation. Location of results: Figure S9.

We also attempted to assess off-target blocking with composite arrays using the plasmid-clearance assay in *E. coli* as a convenient demonstration system. However, the clearance activity for the off-target sites was quite poor, preventing us from advancing these experiments in a reasonable timeframe and drawing robust conclusions about off-target blocking. Instead, we revised our claims about off-target blocking and emphasize that the experiments were only completed *in vitro*. We feel this change has a minimal impact on the overall story, as the experiments were meant to demonstrate merely one of many possible uses of composite arrays.

Reviewer #1 (Remarks to the Author):

In this study, Liao et al. describe a strategy for constructing CRISPR RNA (crRNA) arrays and array libraries, which they use to investigate various aspects of crRNA biogenesis and for multiplex targeting of Cas nucleases. The approach, termed CRATES for CRISPR assembly through trimmed ends of spacers, is a one-pot cloning scheme for crRNA arrays whereby assembly overhangs are positioned within the

trimmed portion of each spacer. CRATES was validated for three Cas nucleases in bacteria and yeast, and used to generate a 125-variant library of three-spacer Cas12a arrays. The authors also generate 'composite' arrays containing two SpCas9 spacer subunits and two FnCas12a subunits, which were used in vitro to reduce off-target cleavage by directing the catalytically inactive FnCas12a to predicted off-target sites of SpCas9. RNA-seq analysis of small RNAs generated by in vitro transcription of a seven-spacer FnCas12a array led to the observation that the terminal repeat generates an unwanted crRNA-like short RNA. The authors suggest that this phenomenon is prevented in native bacterial CRISPR-Cas arrays by the accumulation of mutations in the terminal repeat. Finally, the authors describe a relationship between the secondary structure of transcribed CRATES arrays and the abundance of processed crRNAs. This could be extended to produce a set of design rules to prevent secondary-structure formation in spacer-subunit arrays.

This manuscript raises interesting questions about the biogenesis of crRNAs in native and engineered systems – these insights may prove useful in the design of crRNA arrays for multiplex CRISPR-based applications. However, on its own, CRATES is neither sufficiently innovative in terms of technology development, nor does it address a major technical limitation in the field. (technical limitation in the field: generate large CRISPR arrays efficiently, generate libraries with large CRISPR arrays) Furthermore, with regard to fundamental biological discoveries, the manuscript suffers from several conceptual and technical flaws (detailed below), which cast doubt on many of the insights described by the authors. In particular, most conclusions are based on circumstantial evidence or in vitro / in silico assays, and no experiments are carried out in mammalian cells. Therefore, the manuscript in its present form does not provide sufficient novelty or biological insight to warrant publication in Nature Communications.

We thank the reviewer for their feedback. We address each critique below.

Conceptual issues

1. The paper suffers from a framing issue: the authors place excessive emphasis on CRATES as a novel approach for constructing spacer-subunit arrays. In fact, Gibson assembly and Golden gate cloning are routine techniques that can be used for the assembly of crRNA arrays, and direct synthesis is an effective (and relatively affordable) method for generation of scarless crRNA array libraries. Therefore, all experiments presented in Figures 3-6 would have been possible without CRATES. CRATES is simply a cloning method and as such should not be the main focus of the manuscript – the technique can be described in 1-2 paragraphs in the main text, and more details can be provided in the Methods section. The focus of the paper would seem better placed on the study of crRNA biogenesis and the possible applications of crRNA arrays. In addition, it should also be noted that although CRATES provides a low-cost alternative for assembling array libraries, it does appear to suffer from a very broad distribution of crRNA variant abundance.

We thank the reviewer for their suggestions on how the manuscript could be restructured given our promising insights into crRNA biogenesis. However, we strongly hold that CRISPR arrays remain incredibly difficult to generate via DNA synthesis, Gibson Assembly, or Golden Gate Assembly despite the revolutionary impact these techniques have had on DNA construction, and CRISPR array libraries (to our knowledge) have never been reported. We believe this disconnect may stem from the many advances in assembly sgRNA arrays, which have not been extended to CRISPR arrays. Upon revisiting the introduction of our submitted manuscript, we realized that we did not address how sgRNA arrays are assembled and what this means for CRISPR array assembly. We have now heavily revised the introduction to draw this distinction.

We also wish to fully address the critique raise by the reviewer. Therefore, below we detail the differences between CRISPR arrays and sgRNA arrays and explain what these differences mean for their construction.

CRISPR arrays vs. sgRNA arrays: CRISPR arrays are compact (each subunit contains a ~36 bp conserved repeat and a ~30 bp target-derived spacer) and can be utilized by any associated CRISPR nuclease (including Cas9, Cas12a, and Cas13a). In contrast, sgRNA arrays are much longer (each subunit contains an ~50 – 400 bp promoter-terminator pairs or a ~100-bp cleavage domain in addition to the ~100-bp sgRNA) and generally have only been applied to Cas9. The distinct make-up of the subunits of each array explains why sgRNA arrays (but not CRISPR arrays) could be synthesized or assembled through many of the techniques noted by the reviewer. Each general technique is addressed in turn.

DNA synthesis: many DNA-synthesis companies offer affordable and rapid synthesis of shorter dsDNA “gene fragments”, with the leading product arguably being IDT’s gBlocks. Gene fragments are normally generated by annealing and ligating synthesized oligos, which often fails if the oligos contain repetitive sequences. This is a particular issue for CRISPR arrays given their conserved repeats and short spacers. As one concrete example, attempting to order a two-spacer FnCas12a array lacking the terminal repeat—arguably the simplest, stripped down version of an array one could order—as an IDT gBlock results in the order being rejected for multiple reasons related to the repetitiveness of the sequence (see Figure R1 below). The noted complexities and the total complexity score that resulted in the order being rejected would only increase for larger arrays.

gBlocks® Gene Fragments Entry

Watch a video demo of new features »

Wondering when your order will ship? Check the real time Order Status page! »

BULK INPUT
COLLAPSE ALL
EXPAND ALL

Number of Entries: GO

1

^
🗑

Sequence ⓘ

GTCTAAGAACCTTTAAATAATTTCTACTGTTGTAGATTTGACAGCTAGCTCAGTCCTAGGTATGCTGGT
 CTAAGAACCTTTAAATAATTTCTACTGTTGTAGATACCTCGAGGGGATCCTCTAGATTTAAGAGT

Modifications ⓘ

- 5' Phosphorylation (for blunt cloning only)

TEST COMPLEXITY

Length: 132
 Current base: 133

Denied - High Complexity (Scores of 10 or greater)

The identified complexities prevent manufacturing of this sequence. Click Edit for more information.

Total Complexity Score: 67.4

	Complexity Description	Score
EDIT	A repeat with the sequence GTCTAAGAACCTTTAAATAATTTCTACTGTTGTAGAT exists at the following locations: 67, 1. Solution: Modify the sequence to reduce the length of these repeats to less than 12 bases.	16.8
EDIT	One or more repeated sequences greater than 8 bases comprise 66.7% of the overall sequence. Solution: Redesign to reduce the repeats to be less than 40% of the sequence.	10.7
EDIT	The repeated sequence GTCTAAGAACCTTTAAAT exists at the following locations: 1, 67. Solution: Redesign to disrupt instances of the repeat near the terminal end(s) or add extra bases to push the repeat internally.	10

Figure R1. Screenshot of IDT's gBlock ordering tool following submission of a minimal repeat-spacer-repeat-spacer array for FnCas12a. The conserved repeat is GTCTAAGAACTTTAAATAATTTCTACTGTTGTAGAT. IDT scores the complexity of the sequence according to multiple criteria such as G:C content, length of repetitive sequences, and percentage of the gBlock composed of repetitive sequences. Sequences with total complexity scores of 10 or greater are rejected. The top three identified complexities are shown.

Gene synthesis is another option that relies on a different assembly strategy and therefore avoids some of the issues posed by CRISPR arrays. However, in contrast to gene fragments, gene synthesis is more expensive and requires a longer turnaround time than gene fragments. Furthermore, and more importantly, our consistent experience is that the synthesis of even short CRISPR arrays requires much longer production times, comes with additional service costs, and regularly fails, resulting in cancellation of the order by the company. To this point, IDT's website offers two caveats about gene synthesis directly related to CRISPR arrays (<https://eu.idtdna.com/pages/products/genes-and-gene-fragments/custom-gene-synthesis>):

"Gene sequences with added complexity can interfere with assembly and/or sequencing performance. Such sequences may result in less synthesis yield and/or additional services charges."

"The time required to manufacture a gene is dependent on length, complexity, and vector choice."

As further support, we are aware of only one publication (PMID = 29106617) that relied on gene synthesis to generate the CRISPR arrays used in the work. In this case, the authors ordered one two-spacer array, and one four-spacer array via gene synthesis through the company GeneArt.

We do note that the synthesis of sgRNA arrays can suffer from similar issues, although this issue can be alleviated in part by using dissimilar promoters to introduce large spacing between the remaining conserved Cas9 "handles."

Gibson assembly: this technique can assemble multiple fragments of linear dsDNA through chew-back, hybridization, polymerase extension, and ligation of homologous ends. However, the assembly method breaks down if a common sequence near the DNA ends is present in multiple DNA fragments. This issue would interfere with the assembly of both sgRNA arrays and CRISPR arrays, although sgRNA arrays would be less impacted because the subunits are much larger. Accordingly, we found only three publications (PMID = 25432517, 25917172, 29702666) involving sgRNA arrays assembled through Gibson assembly, without any reports of the assembly efficiency, and all three relied on dissimilar promoters near the assembly site. Instead, Gibson assembly is primarily used when generating libraries with a single sgRNA (e.g. PMID: 28333914).

To our knowledge, no publications have employed Gibson assembly to generate CRISPR arrays, which we again attribute to the closely spaced repeats inherent to each array.

Golden Gate assembly: the final technique relies on defined overhangs generated by Type IIS restriction enzymes, where the annealed overhangs form the assembly junctions between linear fragments of dsDNA. A crux of the technique is selecting highly dissimilar junctions to avoid mispairing and subsequent mis-assembly. Multiple studies have reported validated sets of compatible junction sequences for multi-fragment assemblies (PMID = 27918548, 28976959, 28607761, 29083402). The question is where to place these junctions so they do not interfere with the final construct.

In the case of sgRNA arrays, the assembly junctions are normally placed in the intervening region between the promoter-sgRNA-terminator subunits. Because the intervening regions play no role in

sgRNA expression, no restrictions are placed on the junction sequences. Accordingly, Golden Gate assembly has been the standard strategy for assembling sgRNA arrays, with numerous related techniques and publications.

The story is quite different for CRISPR arrays. Here, it is impractical to clone ~66-bp repeat-spacer subunits prior to array assembly following common Golden Gate assembly schemes. Because of this, and to our knowledge, traditional Golden Gate assembly (i.e. using Type IIS restriction enzymes to generate DNA fragments with overhangs) has never been applied to CRISPR arrays.

More recent work has annealed offset oligos to simulate the cleavage products of Type IIS restriction enzymes. The resulting repeat-spacer subunits can be assembled into an array, offering the only general approach to-date to generate CRISPR arrays. A few prior publications have employed this general approach, where we documented every existing technique that we are aware of in Table S1. As we detail there, these techniques can generate CRISPR arrays, although they all consistently lack the modularity necessary to generate larger CRISPR arrays or CRISPR array libraries. We directly address these limitations with the development of CRATES.

In support of the above justification, an Addgene blog from 2016 (<https://blog.addgene.org/crispr-101-multiplex-expression-of-grnas>) offers a well-described and comprehensive list of all available methods for generating multiplexing gRNA constructs as of that date. Only one on the list (CRISPathBrick) applies to CRISPR arrays, while the nine others only apply to sgRNA arrays. Note that CRISPathBrick was already addressed as part of Table S1.

Above, we addressed how individual CRISPR arrays can be constructed, although we also demonstrated the one-pot construction of CRISPR array libraries. Critically, there have never been any prior publications to our knowledge that constructed or used a library of CRISPR arrays. We do note that there have been multiple reports of constructing and applying libraries of sgRNA arrays (e.g. PMID = 29946130, 30403660, 27238023), although library members never exceeded two guides in an array (e.g. PMID = 26864203). Furthermore, they exploit the assembly techniques described above that cannot be extended to CRISPR array libraries. Given the tremendous potential of performing combinatorial screens using CRISPR arrays, a practical means of generating these particular libraries is therefore needed.

We thus believe that the assembly strategy in itself will be an important advance, particularly for the growing number of researchers using Cas12a and Cas13a and the many researchers that would benefit from combinatorial screens using libraries of CRISPR arrays. The insights gained from characterizing the arrays provides additional support for the significance of our work. We therefore opted to maintain the current structure of the manuscript and our emphasis on the assembly scheme.

The reviewer also raised concerns about the distribution of the arrays in the library we generated. The distribution is actually a boon for the technique, as we only observed a 7-fold difference between the most and least abundant library members. This is a great starting point for any library (e.g. PMID = 29946130 showed a 10-fold difference for their sgRNA library), and we pointed out how the range could be further narrowed by adjusting the levels of the corresponding repeat-spacer subunits.

2. Although the implementation of engineered CRISPR systems in bacteria and yeast is of interest, the majority of current research involving Cas and dCas programmable proteins is focused on engineering

mammalian and plant cells. It is not clear why the authors have only tested CRATES arrays in bacteria and yeast where the practical value is reduced compared to other systems. Although certain dimensions of this study, such as processing of Cas9 crRNA arrays, would be significantly more difficult in eukaryotic cells (since unlike Cas12a and Cas13, Cas9 cannot process its own crRNA arrays), CRATES-generated CRISPR arrays should be tested and the proposed mechanistic insights should be validated in mammalian cells.

We agree that validating our findings in mammalian cells would appeal to the broader audience using CRISPR technologies. We therefore placed the synthetic seven-spacer array assembled using CRATES under a PolII CMV promoter and transiently transfected this plasmid and a plasmid expressing FnCas12a into HEK293T cells. The resulting RNAs were then isolated and subjected to next-generation sequencing. In total, we tested two versions of the array using this approach: one with the consensus repeat and one with the native terminal repeat at the 3' end of the array. These experiments provided multiple insights directly addressing the reviewer's comment:

- **FnCas12a processes the transcribed array into individual gRNAs.**
- **The relative abundance of the gRNAs paralleled those from the cell-free TXTL experiment.**
- **A full-length unintended gRNA was generated from the 3' consensus repeat.**
- **The full-length gRNA was not generated from the 3' native terminal repeat. Interestingly, there was a truncated version comprising only the repeat was present, although this sequence did not harbor the downstream sequence and therefore would not actively target DNA.**

Overall, these results demonstrate that the insights we observed in cell-free systems can directly extended to distinct cellular environments such as those in mammalian cells. The new results are addressed on p. 13, 16-17 and were incorporated into Figures 5 and S10.

3. Figure S3A describes Cas12-inactivating mutations but this analysis does not seem relevant to the theme of the paper. Furthermore, the behaviour of these mutants does not seem to correlate with previous studies. As stated in the manuscript, the D1255A mutant should only partially disrupt catalytic activity – instead, all mutants appear to have the same deleterious effect (the previous study was looking for DNA cleavage and was done *in vitro*, what we did here was investigating gene repression activity and was done *in vivo*). In addition, E1006A shows a significant loss of DNA binding and, surprisingly, additional mutations appear to restore its activity. This effect is very difficult to reconcile and although the authors acknowledge it, they do not offer any explanation for this surprising observation. This leaves the reader to believe that either the story is incomplete or the TXTL assay is unreliable.

As a quick clarification, the repression assay was performed in *E. coli* rather than TXTL. The prior studies we cited were biochemical assays, so some differences are expected when transitioning into an *in vivo* environment. We also recently noticed a publication from November 2018 (Miao et al. *Synthetic and Systems Biotechnology*, 2018), which similarly reported that the mutation of E1006A was compensated by D917A in terms of gene repression by dFnCas12a in *E. coli*. Therefore, the phenomenon was validated independently, even if the exact mechanism remains elusive. We now cite this manuscript on p. 9.

We also purposefully placed this dataset in the supplementary information, as it represented a side observation separate from the major conclusions of our work. We hold that keeping this dataset as part of the work is warranted to inform others that this single point mutation can be problematic

when using Cas12a for programmable DNA binding, especially now that another group made a similar observation. However, given its lesser importance to the overall work, we condensed the description of these mutational analyses in the main text to a single paragraph on p. 8-9.

4. The authors observe unintended crRNA production from terminal repeats in FnCas12a arrays, and suggest that this could titrate the nuclease and lead to off-target DNA damage – however, this phenomenon is only described *in vitro* using the TXTL assay. They infer from sequence analysis that the same would occur in bacteria. However, it is unclear whether this occurs in mammalian cells and, if so, whether errant crRNAs would actually load into Cas12a and lead to mistargeted nuclease activity. Furthermore, the authors should test their assertion that naturally-occurring terminal repeats are mutated to prevent this phenomenon. For example, they could introduce one of these mutations in the last repeat of the synthetic FnCas12a array and assess crRNA generation by RNA-seq. Finally, since removing the terminal repeat does not seem to interfere with the activity of the PcF-1/3/2 array (Figure S6), it suggests that this issue could be simply avoided by creating arrays lacking the last repeat sequence.

We first note that RNA-seq analysis of the native FnCas12a array from *Francisella novicida* (PMID = 26422227) did not identify an unintended crRNA, providing initial *in vivo* evidence. However, we agree that it would be beneficial to demonstrate that the unintended crRNA is active and the native terminal repeat prevents formation of this crRNA in mammalian cells. We therefore repeated the plasmid clearance assays using one of the three-spacer arrays used by FnCas12a (PcF-2/3/1), only the targeted plasmid for clearance encoded the expected target of the unintended crRNA. We also tested two additional arrays in which the 3' repeat was deleted or replaced with the native terminal repeat for FnCas12a. In line with our expectations, the targeted plasmid was efficiently cleared by the three-spacer array with a 3' consensus repeat but not with a native terminal repeat or without a 3' repeat. These results demonstrate that the unintended crRNA identified through RNA-seq analysis can drive efficient DNA targeting. These new results are now addressed on p. 17 and were incorporated as Figure 7B.

We also agree that the 3' repeat could be simply removed, in line with our new experimental results. However, we recently showed that a structured hairpin immediately downstream of this repeat can interfere with crRNA biogenesis and DNA-targeting activity (PMID = 30252595). Notably, the inhibitory effect was exacerbated when the 3' repeat was deleted, potentially complicating this otherwise simple strategy. In this same work, we showed that inhibition could be alleviated by replacing the 3' repeat with a native terminal repeat. This point is made with the following text on p. 19 of the discussion:

“[T]he terminal repeat was shown to be important when deriving Cas12a crRNAs from eukaryotic mRNAs, and we recently showed that processing of the transcribed FnCas12a array was sensitive to adjacent stable secondary structures that was exacerbated in the absence of a terminal repeat.”

Technical concerns

1. Although the authors describe their quantification analysis, it is impossible to infer error or significance from heat maps. The heatmaps should be accompanied by conventional plots containing error bars and appropriate statistical tests.

We now provide bar graphs with error bars of all main-text heat maps in the SI, which further support the main conclusions we draw. These plots are shown in Figure S2. We also added statistical analyses where appropriate. We find the heat maps are a more digestible way to communicate these large data sets.

2. The Sanger sequencing data supporting the claim that assembly scars are trimmed during processing is not provided (this result is hidden in NGS data in Figure 5). Since this assumption lies at the foundation of CRATES, this data should be provided in the supplementary figures.

We are not sure how Sanger sequencing would be helpful, as we are evaluating a diverse set of RNA sequences. We do note that spacer trimming is well established in the literature dating back to the original reports of crRNA biogenesis by Cas9 (PMID = 21455174), Cas12a (PMID = 26422227), and Cas13a (PMID = 27256883).

A related and important point is that numerous studies have shown that the actual region employed for target recognition is limited to the first 20 – 24 PAM-proximal nts in the guide, while our junctions are placed well outside of this region. Therefore, even a partially trimmed RNA would not utilize the junction sequence as part of target recognition. As a result, it is inconsequential whether the junction is fully trimmed in every crRNA. We have made revisions throughout the manuscript to convey this point. For instance, we have added the following to p. 4:

“CRATES takes advantage of the portion of crRNA spacers that does not participate in target recognition and often undergoes trimming as part of crRNA biogenesis for single-effector Cas nucleases.”

Or the following on p. 10:

“Next, we asked if CRATES could be extended to other CRISPR single-effector nuclease, given that the associated crRNA spacers also undergo trimming and do not rely on the full spacer for target recognition.”

3. Sequentially delivering dFnCas12a and SpCas9 to a cell-free TXTL system is not an appropriate method for assessing nuclease protection in a cellular environment (Figures 4D, S4A-C). These experiments should be performed in cells (ideally in mammalian cells) using a competition assay whereby different fluorescent reporters are coupled to the on-target and off-target sites, and the activity of SpCas9 is assessed in parallel at each site in the presence or absence of dFnCas12a.

This experiment was meant to highlight merely one of many different applications of composite arrays. We also showed directly that, aside from off-target blocking, the composite arrays can be used for coordinated plasmid clearance and gene repression in *E. coli*. That said, we did try assessing nuclease protection using the plasmid clearance assay in *E. coli* as a reasonable test given the time constraints of the revision process. However, after developing the associated constructs, we found that the plasmids with the off-target sites were poorly cleared by Cas9, confounding our ability to reasonably assess nuclease protection in a cellular environment. We therefore opted to modify the text to downplay this specific application of the composite arrays, including condensing the relevant section in the Results and deleting the section in the discussion.

4. The analysis of crRNA secondary structure relies entirely on output generated by the predictive algorithm NUPACK. Since RNA secondary structure varies between species and cell types due to the cellular and protein environment, *in silico* modelling is only predictive of a certain fold. Therefore, it is difficult to extrapolate results from *in silico* predictions into bacteria or mammalian cells. In addition, the analysis is further weakened by the lack of proper randomised controls to establish the baseline folding capacity of similar length RNA sequences. For example, the authors should randomly shuffle the array sequence (thousands of iterations) and predict folding *in silico* to determine the background mean and standard deviation contribution to secondary structure at each nucleotide position. Finally, these predictions should be tested in cells by constructing different synthetic arrays and comparing their activity (as in Figure 6E).

We appreciate the reviewer's desire for a fully systematic analysis of how RNA secondary structures contribute to crRNA biogenesis and DNA targeting, particularly given concerns around deviations from the *in-silico* predictions. However, NUPACK and other secondary-structure prediction algorithms have been widely and successfully used for predicting targeting by bacterial small RNAs (PMID = 18399940, 21742981, 22388518) as well as guiding the design of synthetic RNAs that function *in vivo*, including for the design of toe-hold switches (PMID = 25417166), base-pairing RNAs (PMID = 24833802), transcriptional terminators (PMID = 21555549), translational repressors (PMID = 22446835), ribosome binding sites (PMID = 19801975), transcriptional activators (PMID = 25643173), riboswitches (PMID = 17709748, 18956013, 26621913), sgRNAs (PMID = 26824432), and others. We acknowledge that these algorithms are not perfect and miss known interactions such as pseudoknots, triple helices, and non-canonical base pairing as alluded to by the reviewer. However, they often provide useful predictions. Also note that we provided a total of four examples in which the predicted secondary structure correlated with crRNA abundance: the synthetic seven-spacer array assessed in TXTL, two synthetic three-spacer arrays assessed in TXTL and in *E. coli*, and the natural nine-spacer array that was previously assessed in *F. novicida*. This set already represents a larger analysis.

Furthermore, we provided *in vivo*, mutational data supporting the role of the predicted structure in inhibiting crRNA-directed plasmid clearance (Figure 6D-E). The data consistently yielded the expected results assuming a role for secondary structure: five mutations that disrupt the predicted structure improved plasmid clearance, whereas two further mutations that restore the predicted structure resulted in reduced plasmid clearance. We also provided new data as part of the revision process (Figures 5, S10) showing the pattern of gRNA abundance was similar when the associated CRISPR array was expressed in TXTL and in mammalian cells.

Finally, we believe that providing multiple lines of evidence of the potential influence of global secondary structure to crRNA abundance and targeting ability is an important advance given that most have assumed that crRNA abundance is principally influenced by the distance from the promoter. We see the systematic analysis suggested by the reviewer as the next important step—albeit beyond the scope of this work. Instead, this work sought to demonstrate the novel assembly technique and provide demonstrations on how the technique can be applied to achieve multiplexing and interrogate the properties of CRISPR arrays and crRNA biogenesis.

5. The claim regarding context-dependent loss of crRNA-directed FnCas12a activity is only supported by the analysis of two arrays – the single array that displayed poor plasmid clearance activity (PcF-2/3/1), and one that was very effective (PcF-1/3/2). To establish whether the imperfect global secondary

structure of the array (i.e. disruption of the repeat hairpin) is responsible for this phenomenon, these analyses should be carried out using the other four array combinations that were effective at plasmid clearance. To substantiate this conclusion, it is crucial to determine spacer production from the PcF 1/2/3, 2/1/3, 3/1/2, and 3/2/1 arrays, and compare the secondary structure of various hairpins across the arrays. In addition, since the band intensities in the Northern blot are very weak, replicate blots should be provided as supplemental figures and band intensities should be quantified for statistical analysis.

In line with this reviewer's request, we have added the folding predictions for the other four three-spacer arrays as part of Figure S9. These analyses show that only one array harbors a predicted stable secondary structure that would be expected to impact crRNA processing.

We have also included an independent replicate experiment of the Northern blotting to reproduce the reduced abundance of the processed crRNA, as requested by this reviewer. The resulting image was incorporated as Figure 6C, while the image from the previous Northern blotting analysis is now shown in Figure S7. In our experience, Northern blotting analysis of transcribed CRISPR arrays in TXTL and *E. coli* yields the processed gRNA as the minor product (e.g. see PMID = 30252595). We feel that the gel images are sufficient without further analyses, as they are meant to confirm the RNA-sequencing results. Note that these analyses were also conducted with a separate three-spacer array to show again that the *lacZ* promoter-targeting spacer was lower only for PcF-2/3/1.

Even if the extensive comparative analysis was not performed, we did provide extensive mutational analyses that strongly supported the underlying mechanism for the lost plasmid-clearance activity. These analyses included disrupting as well as reforming the predicted imperfect hairpin, providing experimental evidence for the contribution of RNA secondary structure.

Minor issues

1. Panel 2D is not referenced in the text – it should be referenced on line 128.

We thank the reviewer for catching this error. We have added the figure citation on p. 8 in the revised manuscript.

2. Line 140 should read “AsCas12a”.

We again thank the reviewer for catching this error. “FnCas12a” has been replaced with “AsCas12a.”

3. Missing reference in line 182.

While condensing this section based on the reviewer's recommendation, the related sentence was deleted.

4. Line 203 states that the dFnCas12a double mutant was used for the VP64 fusion – in Figure 3C and the associated legend it is stated that the triple mutant was used. The authors should clarify this.

We thank the reviewer for catching this otherwise easy-to-miss error. Figure S3C has been updated to indicate the correct number of white dots, and the corresponding legend has been updated to indicate whether a double or triple mutant was used.

5. Figure 3C –spacers P_cL-nt and P_cL-1 clearly display non-specific activity (in particular at PtL3 target) – this is ignored in the text. The authors should explain this phenomenon and provide the raw data and associated quantification.

As suggested by this reviewer, we have included bar graphs with error bars in Figure S2C. When analyzing the data following the reviewer’s observations, we realized that the apparent non-specific activity can be mostly attributed to variability in the final GFP levels for each TXTL run—a common occurrence when applying TXTL. However, normalizing GFP values to that of the non-targeted construct (PtL-nt) for each array reduced this variability. PtL3 still yielded slightly lower GFP levels when paired with P_cL-1 and P_cL-2, although we felt that the effect was sufficiently minor to not warrant a comment in the text.

We have updated Figure 2C, the figure legend, and the Methods section to reflect this change in data representation.

6. Flow cytometry gating strategies for yeast experiments should be included as supplemental data.

As suggested by this reviewer, we have included a representative example of the gates used to identify events containing yeast cells. These images were incorporated as Figure S3D.

7. The variant library cloning scheme is not explained in sufficient detail.

As suggested by this reviewer, the description of the cloning scheme in the Methods section as well as the SI protocol has been expanded. For instance, we have added the following to p. 29:

“As part of the library design, the 5’ and 3’ assembly junction sequences were used for all repeat-spacer pairs destined for the same position in the array (e.g. spacers S1a-e destined for the 5’ position in the three-spacer array included annealed oligos with a 5’ CCCT overhang at the 5’ end and a 5’ GCTG overhang at the 3’ end). Otherwise, the oligo design followed exactly that specified under “One-pot generation of CRISPR arrays.” Table S4 contains the specific oligonucleotides and assembly junctions used for the library generation.”

8. A positive readout for measuring the efficiency of CRATES arrays (such as flow cytometry using a fluorescence reporter) would be more reliable than a negative readout (i.e. loss of antibiotic resistance).

We agree positive readouts are more reliable than negative readouts, although the plasmid clearance assay resulted in multiple orders-of-magnitude drop in the transformation efficiency. The bar graphs and error bars displayed in Figure S2 show that the output was quite robust.

9. The authors should clarify that the experiments in Figure 3B and 4 rely on the host bacterial nucleases to process the Cas9 spacer-subunit array.

As suggested by this reviewer, we have updated both descriptions to state that we relied on RNase III endogenous to *E. coli*. For instance, we added the following to p. 10 when describing the experiment in Figure 3B:

“We also encoded the tracrRNA required for crRNA processing as part of the Cas9 expression plasmid, and we relied on the RNase III endogenous to *E. coli*.”

10. Figure legend S4 should read “see Figure 4E” instead of “see Figure 5E”.

We thank the reviewer for catching this error. The reference has been updated accordingly.

11. In Figure 6B, the authors compare arrays PcF-2/3/1 and PcF-1/3/2; yet, in Figure 6C, they switch the effective array to PcF-3/2/1, without providing any explanation or mentioning this in the text. As discussed above, all these analyses (RNA-seq, Northern blot and RNA secondary structure prediction) should be carried out for all array combinations.

As recommended by the reviewer, we have added the following text to p. 15 to explain why the different arrays were used for RNA-seq analysis and Northern blotting:

“We also evaluated an array in which spacer S1 and a different spacer were switched (PcF-1/3/2) to distinguish between the contributions of the spacer and the spacer position.”

“As further confirmation, we performed Northern blotting analysis to detect the crRNA derived from spacer S1 in *E. coli* expressing FnCas12a and the arrays from PcF-2/3/1 or PcF-3/2/1, where the first two spacers were switched. This analysis showed that the crRNA derived from spacer S1 was less abundant for PcF-2/3/1 than for PcF-3/2/1 (Figs. 6C, S7).”

We also agree performing the suggested analyses across all six arrays would provide a more comprehensive picture. However, we felt that the explanation above combined with the folding predictions (Figure S9) and the mutational analyses (Figure 6D-E) are sufficient to support our conclusion that secondary structure can impact crRNA abundance and targeting activity.

Reviewer #2 (Remarks to the Author):

Liao and co-workers describe the development of a one-pot assembly approach for CRISPR arrays. This straightforward method appears to simplify the generation of design arrays for multiplexing by CRISPR associated nucleases, or even sets of these enzymes. Apart from the CRISPR assembly, this study describes a systematic investigation of the functionality of CRISPR-encoded guides, revealing the influence of context in the precursor-crRNA, most likely reflecting an important role of secondary structure. This paper seems well executed, and very well described and illustrated. The reported information will be of interest for the broad genome editing community. There are some issues that need attention.

Comments

1. In lines 27- 29 it is mentioned “joining sgRNAs with intervening cleavage domains to form sgRNA arrays”. It would be appropriate to cite Ferreira et al. ACS Synth Biol (2018).

We now cite this article on p. 3 as recommended by the reviewer.

2. Line 31-33: “yet the requirement for a tracrRNA and RNase III limited the use of CRISPR arrays utilized by Cas9 to bacterial applications”. This sentence should be adjusted. There are many examples in the literature of eukaryotic organisms which make use of the CRISPR arrays utilized by Cas9 with tracrRNA and endogenous RNase III. See Cong et al. Science (2013), as well as Mali et al. Science (2013) . Also, it should be mentioned that a synthetic single guide does not require RNaseIII (Jinek et al. Science (2012)).

While revising the introduction to address comments from Reviewer #1, we deleted the sentence about CRISPR arrays and bacterial applications. We have also rephrased the sentence about the requirements for a tracrRNA or RNase III on p. 3 to the following so it’s more clearly that neither factor is required when using sgRNAs:

“These sgRNAs circumvented the need for either a tracrRNA or RNase III that both participate in crRNA biogenesis.”

3. Line 34: should read “process precursor-crRNAs”

We have added precursor CRISPR RNAs as part of our initial description of CRISPR arrays and crRNA biogenesis on p. 3. Otherwise, the sentence noted by the reviewer was deleted while address comments from Reviewer #1.

4. Line 36: would be appropriate to add Reference of 4-gene multiplexing in yeast: Swiat et al. NAR (2017)).

We have added the citation on p. 3 as recommended by the reviewer.

5. Line 44: typo - 'for even for' > 'even for'

We thank the reviewer for catching this typo. We note that the corresponding was deleted as part of addressing comments from Reviewer #1.

6. Figure 1B: not all repeat-spacer subunits are designed in the same way, meaning that some have 3' and 5' overhangs while others only 5' overhangs. This does not match with what is stated in lines 90 and 91: "Repeat-spacer subunits were designed to contain alternating 5' and 3' overhangs"

As suggested by this reviewer, we have rephrased this sentence on p. 6 to read the following:

"Repeat-spacer subunits were designed to contain 5' and/or 3' overhangs with highly dissimilar sequences previously validated for efficient modular assembly..."

7. Line 110: it is mentioned that arrays containing 7 spacers were correct 60% of the times. Did you sequence the false positive colonies - are they spontaneous deletions of gfp? Does the efficiency drop because of the chosen 4 bp linkers or because of the length of the array? Please comment.

We did not sequence these false positive clones as part of the cloning experiments, although the lack of fluorescence of the colony and the smaller resulting PCR product (Figure 1C) indicate that the arrays had misassembled, forming an array with only a single spacer. We have added the following to p. 7 to capture this point:

"In the case of the two negative clones, the smaller PCR products were in line with formation of arrays with a single spacer."

Concerning the efficiency drop, we currently attribute this to trying to assemble more pieces at one time, which would be associated with lower efficiencies on its own. We therefore have added the following to p. 6-7:

"We found that the total number of transformants decreased for arrays with more spacers, in line with lower efficiencies when assembling more fragments at one time."

8. References 13 and 31 are the same.

We thank the reviewer for catching this error with our citation program. We have ensured that ref. 13 does not appear again. We have also reviewed the references to confirm no other duplicates are present.

9. Line 139-142: Clarify if this is a different experiment from the one described in lines 142-148. Was this first experiment done with FnCas12a or AsCas12a?

We thank the reviewer for catching this error, where we incorrectly stated FnCas12a rather than AsCas12a in this sentence. We have replaced "FnCas12a" with "AsCas12a."

10. Lines 181-182: “Cas12a nucleases can process transcribed CRISPR arrays in eukaryotic cells without accessory factors”. This feature is not exclusive for eukaryotic cells; please rephrase.

This sentence was deleted as part of the revisions made to address comments by Reviewer #1.

11. Line 187: since this experiment is carried out in yeast, it should be mentioned that gene regulation has also been reported using the dead version of LbCas12a in yeast combined with different activators (VP64 (V), VP64-p65AD (VP), and VP64-p65AD-Rta (VPR)) (Lian et al. Nat Commun (2017)). This is also applicable for justifying lines 202- 204.

Following the reviewer’s recommendation, we added this citation to p. 9 when introducing the gene-activation experiments in yeast.

12. Lines 243-245: State the importance of this finding and compare it with other research. The swapping was only done for Cas9 spacers. Worth to include an extra experiment in which Cas12a spacers are swapped. In the experiment where the blocking of off-target sites is done, you swap the order but in this way, you could explore whether the context of the spacers can impact the resulting activity and if this phenomenon is sequence specific.

The two spacers used with FnCas12a in these experiments are also used in Figure 6A (S1 and S2). We were already aware that the S2 spacer proceeding the S1 spacer could disrupt the targeting activity of S1, so we opted not to explore this in lieu of the work shown in Figure 6. We have instead added the following to p. 12 to capture the novelty of this insight and suggest that the effect could also be sequence-dependent:

“...suggesting that the sequence and context of the spacers in an array can impact the resulting crRNA-directed nuclease activity.”

13. Lines 319-328: “CRATES revealed context-dependent loss of crRNA-directed FnCas12a activity”. It is context dependent but also sequence dependent. For this reason, this result does not match with previous results that state that the context of the spacer in the array does not affect its activity. Please rephrase.

We agree that the loss of targeting activity can also be considered sequence-specific since this was only observed for spacer S1. We have modified the heading and experimental conclusions on p. 14 accordingly. For instance, the heading now reads the following:

“Using CRATES revealed sequence- and context-dependent loss of crRNA-directed FnCas12a activity related to secondary structure formation.”

We also agree that this conclusion conflicts with at least one prior study (Zetsche et. al Nat Biotechnol 2017) stating that the spacer order does not matter. We therefore have noted this difference and how we resolve these conflicting conclusions in the Discussion on p. 19:

“[P]rior work did not observe any effect on spacer order, although the number of tested arrays and spacer sequences has been limited.”

14. All the results state: “CRATES ...”. CRATES is the assembling method. All the results that are obtained do not have to do with the assembling method itself but are observations of the activity of the arrays... . Please rephrase.

We agree our original phrasing was imprecise. We have therefore corrected this in every instance in the manuscript, including the Abstract, Results, and Discussion. For instance, the heading on p. 14 now reads:

“Using CRATES revealed sequence- and context-dependent loss of crRNA-directed FnCas12a activity related to secondary structure formation.”

15. Discussion: it is important that it is stated that the results are spacer-sequence dependent (as indicated in the end of the discussion when talking about future design tools) . Please rephrase.

We agree it’s important to note that the effect is also sequence-dependent. We therefore have slightly revised the abstract and added the following to Discussion on p. 18-19:

“...suggesting a dependence on the sequence of the spacer and its context in the array.”

16. Line 688 of methods: “in SOC media” is repeated. Please rephrase.

We thank the reviewer for catching this error. The second instance of “in SOC media” was deleted.

Reviewers' Comments:

Reviewer #1:

Remarks to the Author:

The authors have revised the text and figures of their manuscript and conducted additional experiments. However, some important issues remain regarding the functionality of CRATES and the proposed biological insights.

Major points

1. The authors have convincingly argued that the convenience and versatility of CRATES as a strategy for assembling CRISPR arrays represents an improvement over existing methods. However, this reviewer maintains that CRATES does not constitute a major technological breakthrough in itself. As the authors have acknowledged, although each method has its advantages and limitations, other means for the assembly of CRISPR arrays have been reported, including chemical synthesis (PMID: 29106617) and methods based on Type IIS restriction enzymes (Golden Gate assembly, CRISPathBrick).

2. The authors argue that the 7-fold difference observed in the distribution of their arrays is not a concern. To support this claim, they cite a previous manuscript that reported 10-fold differences between sgRNAs within a much larger library (~60,000 sgRNAs) (PMID: 29946130). Indeed, the standard in the field for library uniformity is <10-fold between the highest and lowest expressed sgRNA. However, this refers to libraries of tens of thousands of sgRNAs. Although 7-fold is within this accepted range, it remains unclear if this distribution will be maintained when scaling up CRATES to thousands or tens of thousands of gRNAs.

3. In response to reviewer comments, the authors have transiently expressed FnCas12a and a CRATES array driven by a CMV promoter in HEK293T cells. They use RNAseq to assess whether the array is processed into individual crRNAs, but they do not show that the crRNAs are active in these cells. It would be simple to demonstrate activity using a fluorescent reporter system – this is an essential experiment to determine the functionality of CRATES in mammalian cells, and it should be carried out for each crRNA in the array.

In addition, the analysis of seven-spacer arrays in HEK293T cells raises further doubts about the performance of CRATES. For example, in this context some spacers appear not to be expressed at all - see spacers 3 and 5 in Fig. S10 and spacer 5 in Fig. 5. Furthermore, some repeat regions are expressed more than spacers. In particular, repeat sequences towards the 3' end of the array appear to be processed much less efficiently in HEK293T compared to the TXTL assay or the 9-spacer array from Zetsche et al (see repeat 7 in Fig. 5 and Fig. S10). How are these observations explained and how do they impact the utility of CRATES?

4. The authors have conducted additional plasmid clearance assays to show that the unintended 3' crRNA is active in *E. coli* (Fig. 7B). They should include an equivalent functional assay for testing the activity of this errant crRNA in HEK293T cells.

5. The analysis of the terminal repeat in HEK293T cells raises additional concerns. In this instance, the native terminal repeat is the most abundant species in the entire array – this is in stark contrast to the native array from Zetsche et al (Fig. S6c) where the terminal repeat is not present at all (i.e. correctly processed?). The authors acknowledge this in line 376 but do not provide any explanation for this observation. In the absence of functional assays in HEK293T cells, it is very difficult to conclude that the native terminal repeat in this case eliminates the generation of unintended crRNAs. For example, reads are visible beyond this repeat, at frequencies that are higher than some of the intended spacers (such as crRNAs 3 and 5 in Fig. S10). Thus, it is possible that a functional errant crRNA is generated from this CRATES array containing a native terminal repeat in HEK cells. This would significantly call into question whether the authors' insights

regarding terminal repeats can be applied to mammalian cells.

Based on these considerations, the conclusion that “[...] expressing the seven-spacer array with the native terminal repeat from *F. novicida* in HEK293T cells eliminated the full-length unintended crRNA [...] (Fig. S10)” is not supported by experimental evidence. Since the proposed role of native terminal repeats is one of the biological insights of the study, further experiments will be necessary to unequivocally prove the validity of this claim.

6. In the revised manuscript, the authors downplayed the importance of spacer trimming by arguing that CRATES assembly scars are located in a portion of the crRNA spacers which does not hybridise to the target sequence. It is still important to determine whether or not these assembly junctions are present in the mature crRNAs because they could affect the specificity and efficiency of targeting. The RNAseq data (Fig. 5) suggests that spacer trimming is incomplete, in particular in mammalian cells: the authors should use this data to determine the true extent of spacer trimming.

7. The authors have included graphs with error bars to supplement their heat maps. They have not, however, added any statistical analysis, without which it is difficult to infer whether the differences observed are significant.

Minor points

- The legend for Figure S9 should read “approach”.
- The sentence on lines 191-2 is very confusing.
- Line 193: should read “FnCas12a”
- Line 210: “nuclease” should be plural.
- Line 88: “undergo” should be singular.

Reviewer #2:

Remarks to the Author:

All matters raised by me have been dealt with appropriately.

We thank both reviewers for their second round of feedback. We provide point-by-point responses below in red. Corresponding changes to the revised text are also in red. Please note that we have changed the nomenclature “unintended crRNA” to “extraneous crRNA” in the manuscript.

Reviewers' comments:

Reviewer #1 (Remarks to the Author):

The authors have revised the text and figures of their manuscript and conducted additional experiments. However, some important issues remain regarding the functionality of CRATES and the proposed biological insights.

We thank the reviewer for thoroughly reading our revised manuscript and for their efforts to further strengthen our manuscript.

Major points

1. The authors have convincingly argued that the convenience and versatility of CRATES as a strategy for assembling CRISPR arrays represents an improvement over existing methods. However, this reviewer maintains that CRATES does not constitute a major technological breakthrough in itself. As the authors have acknowledged, although each method has its advantages and limitations, other means for the assembly of CRISPR arrays have been reported, including chemical synthesis (PMID: 29106617) and methods based on Type IIS restriction enzymes (Golden Gate assembly, CRISPathBrick).

We also maintain that CRATES represents a major advance for the generation of CRISPR arrays, particularly over the few prior methods. Unlike these prior methods, CRATES is the first fully modular assembly strategy by decoupling the assembly junctions from the repeats and spacers. In addition, it allowed—for the first time—the generation of CRISPR array libraries. We believe the demonstration of one-pot library assembly in particular is sufficient to justify the significance of the technique. We also point out that, through the course of this one study, we generated a total of 188 arrays (including the library). This number dwarfs the number of CRISPR arrays generated in prior studies by at least an order-of-magnitude, further underscoring the power of the method.

We also hold that the other methods that have been used to generate CRISPR arrays (and noted by the reviewer) possess major drawbacks that underscored the need to develop CRATES. For instance, for chemical synthesis, we again note that we and others have encountered major issues when ordering arrays through standard suppliers (e.g. IDT). It's therefore telling that we could only find one study (noted by the reviewer) that relied on chemical synthesis to generate arrays. Separately, for CRISPathBrick, this methodology requires a cloning step for each added repeat-spacer, creating a major burden when generating larger arrays. There were also only a single follow-up publication using the technique (from the same group), highlighting potential limitations with its use. The reviewer does note potential issues below that they see as constraints to CRATES, although (as we argue below) these constraints pertain to CRISPR arrays in general versus any one assembly scheme. Therefore, we strongly hold that CRATES represents a major technical advance over other prior means to generate CRISPR arrays.

2. The authors argue that the 7-fold difference observed in the distribution of their arrays is not a concern. To support this claim, they cite a previous manuscript that reported 10-fold differences between sgRNAs within a much larger library (~60,000 sgRNAs) (PMID: 29946130). Indeed, the standard in the field for library uniformity is <10-fold between the highest and lowest expressed sgRNA. However, this refers to libraries of tens of thousands of sgRNAs. Although 7-fold is within this accepted range, it remains unclear if this distribution will be maintained when scaling up CRATES to thousands or tens of thousands of gRNAs.

Upon review, our comparison gave too much credit to the prior report (PMID = 29946130). Instead, below is the original context of the 10-fold range we cited in the publication:

“Moreover, the relative abundance of the majority of sgRNAs (>80%) and genes (median sgRNA read count, >93%) is kept within 10-fold range (Supplementary Fig. 6), supporting the distribution uniformity of the synthetic library.”

If we apply the same cut-off (80% of the gRNA library), then the distribution is within a 3.2-fold range. We now state this smaller range and cite the prior publication in the revised Discussion, and we note that further optimization may be needed when scaling to larger libraries on page 18.

3. In response to reviewer comments, the authors have transiently expressed FnCas12a and a CRATES array driven by a CMV promoter in HEK293T cells. They use RNAseq to assess whether the array is processed into individual crRNAs, but they do not show that the crRNAs are active in these cells. It would be simple to demonstrate activity using a fluorescent reporter system – this is an essential experiment to determine the functionality of CRATES in mammalian cells, and it should be carried out for each crRNA in the array.

We made the decision not to evaluate whether the assembled arrays could direct nuclease activity in mammalian cells for two important and related reasons: (1) the final products of CRATES (as well as other assembly techniques) are CRISPR arrays, and (2) CRISPR arrays have been successfully used in numerous studies for editing and gene regulation in mammalian cells and plant cells (e.g. PMID = 27918548, 29083402). We therefore believe that such a demonstration would add little to the existing body of literature and the overall significance of our manuscript. We also demonstrated that CRISPR arrays generated through CRATES are functional in TXTL, *E. coli*, and yeast, providing ample evidence for functionality in different cellular contexts. Instead, the major advance was demonstrating the variability in crRNA abundance in mammalian cells, that this variability trended with what we observed in TXTL, and that an unintended crRNA was generated.

Prior examples that used CRISPR arrays in mammalian cells and plant cells are already cited on p. 3 of the Introduction.

In addition, the analysis of seven-spacer arrays in HEK293T cells raises further doubts about the performance of CRATES. For example, in this context some spacers appear not to be expressed at all - see spacers 3 and 5 in Fig. S10 and spacer 5 in Fig. 5. Furthermore, some repeat regions are expressed more than spacers. In particular, repeat sequences towards the 3' end of the array appear to be processed much less efficiently in HEK293T compared to the TXTL assay or the 9-spacer array from

Zetsche et al (see repeat 7 in Fig. 5 and Fig. S10). How are these observations explained and how do they impact the utility of CRATES?

We must emphasize that the low abundance of crRNAs noted by the reviewer has little to do with CRATES as a method but instead is associated with CRISPR arrays in general. This point is highlighted by the wide variability in crRNA abundance associated with native CRISPR arrays in bacteria (PMID =21455174, 26422227, 29499139), including the CRISPR array from *Francisella novicida*. We also note that RNA-seq analyses have been rarely applied to synthetic CRISPR arrays, likely leading others to overlook crRNA abundance as a contributing factor to targeting activity. To that point, we see the variability in crRNA abundance as a critical insight into CRISPR array design that so-far has not been reasonably considered by the technology side of the field.

We agree that there were some differences in the crRNA profile between the transcribed arrays in HEK293T cells versus in TXTL, such as incomplete processing near the 3' end of the array in HEK293T cells. We attribute these differences to factors distinguishing the systems, such as the collection of active ribonucleases as well as transient transfection versus cell-free expression. Any differences again would reflect the use of CRISPR arrays rather than the use of CRATES to assemble the arrays. To indicate this difference, we have added the following to the Results section on p. 14:

“There were some differences in the final profile of crRNAs generated in TXTL versus HEK293T cells (e.g. different extents of processing), which likely reflects the associated cohorts of active RNases as well as cell-free versus transient expression. However, the relative abundance of the crRNAs was similar between TXTL and mammalian cells—supporting the role of secondary structure in crRNA abundance independent of the cellular context.”

4. The authors have conducted additional plasmid clearance assays to show that the unintended 3' crRNA is active in *E. coli* (Fig. 7B). They should include an equivalent functional assay for testing the activity of this errant crRNA in HEK293T cells.

We would agree with the reviewer's recommendation if unintended crRNAs were the primary focus of this manuscript. However, given that the manuscript includes other major demonstrations and findings (e.g. the first modular one-pot assembly scheme for CRISPR arrays; first assembly of CRISPR array libraries; demonstration of composite arrays; and confirmed role of global secondary structure in crRNA abundance and targeting activity), we therefore hold that the discovery of unintended crRNAs from Cas12a arrays and the potent activity of these crRNAs in *E. coli* is sufficient for an initial demonstration.

To ensure we do not overstate our findings, we have added the following to the Discussion on p. 18:

“The more practical ramification is that the terminal repeat-derived crRNA could lead to unintended targeting, at least in *E. coli*. Future work should evaluate the activity of these crRNAs in other cellular contexts and whether a native terminal repeat or no terminal repeat better promotes multiplexed targeting by an array.”

5. The analysis of the terminal repeat in HEK293T cells raises additional concerns. In this instance, the native terminal repeat is the most abundant species in the entire array – this is in stark contrast to the

native array from Zetsche et al (Fig. S6c) where the terminal repeat is not present at all (i.e. correctly processed?). The authors acknowledge this in line 376 but do not provide any explanation for this observation. In the absence of functional assays in HEK293T cells, it is very difficult to conclude that the native terminal repeat in this case eliminates the generation of unintended crRNAs. For example, reads are visible beyond this repeat, at frequencies that are higher than some of the intended spacers (such as crRNAs 3 and 5 in Fig. S10). Thus, it is possible that a functional errant crRNA is generated from this CRATES array containing a native terminal repeat in HEK cells. This would significantly call into question whether the authors' insights regarding terminal repeats can be applied to mammalian cells.

Based on these considerations, the conclusion that “[...] expressing the seven-spacer array with the native terminal repeat from *F. novicida* in HEK293T cells eliminated the full-length unintended crRNA [...] (Fig. S10)” is not supported by experimental evidence. Since the proposed role of native terminal repeats is one of the biological insights of the study, further experiments will be necessary to unequivocally prove the validity of this claim.

We hold that the quoted statement is supported by the experimental evidence, as virtually all of the accumulated RNA product covers only the native terminal repeat, and less than 2% of the reads (48/3116) have more than 6 nt of the downstream sequence at the 3' end. Even for the very few reads with an RNA containing a longer downstream sequence, these RNAs would not serve as functional gRNAs since the stem loop for recognition by Cas12a is missing (PMID: 26422227). We have therefore added to Figure S10 to show all the related individual reads and alignment to the array. We have also briefly expanded our description in the Results section to capture this further analysis.

We do agree with the reviewer that the accumulated repeat is at odds with the RNA-seq analysis of the native CRISPR array from *Francisella*. The most reasonable explanation is that the product formed as part of crRNA biogenesis is stable in HEK293T cells, reflecting a common difference in the cohort of active ribonucleases in bacteria versus in mammalian cells. We have therefore revised this sentence in the Results section to the following to better address the unique findings from RNA-seq on p. 16:

“We also found that expressing the seven-spacer array with the native terminal repeat from *F. novicida* in HEK293T cells eliminated the full-length unintended crRNA (Figure S10), although there was a standalone product of ~26 nts representing the terminal repeat with ~10 nts trimmed from its 3' end. This product is unlikely to bind to Cas12a given the lack of a formed stem loop structure recognized by the nuclease. The absence of the standalone product in *Francisella* (Figure S6) also likely reflects differences in RNA stability between bacteria and mammalian cells.”

One alternative is eliminating the terminal repeat. But it will bring the risk of potential inhibitory effect as we noted previously in the Discussion on page 19.

6. In the revised manuscript, the authors downplayed the importance of spacer trimming by arguing that CRATES assembly scars are located in a portion of the crRNA spacers which does not hybridise to the target sequence. It is still important to determine whether or not these assembly junctions are present in the mature crRNAs because they could affect the specificity and efficiency of targeting. The RNAseq data (Fig. 5) suggests that spacer trimming is incomplete, in particular in mammalian cells: the authors should use this data to determine the true extent of spacer trimming.

To the reviewer's point, we agree that users of CRATES should be aware that trimming may not be complete and could yield a crRNA containing a portion of the junction. While the data for characterized Cas9 and Cas12a nucleases clearly show that the junction falls well outside of the region involved in target hybridization (e.g. hybridization ends with nt 20, while the junction begins with nt 27), this may not be the case for other nucleases. We have therefore noted the following on p. 13:

"We also noted that a portion of the crRNAs included some of the junctions, potentially affecting targeting if this region is involved in target hybridization for other CRISPR nucleases."

7. The authors have included graphs with error bars to supplement their heat maps. They have not, however, added any statistical analysis, without which it is difficult to infer whether the differences observed are significant.

We had added statistical analyses in the main text (P. 15) where we felt a statistical comparison was necessary based on the reviewer's original comment. As a further extension, we have added statistical analyses to further support the conclusions drawn regarding Figure 6E (P. 15) and Figure 7B (P. 17).

We also decided not to add statistical analyses to Figure S2, as the data was almost entirely used to conclude whether plasmid clearance occurred, and prior studies reporting plasmid clearance also do not perform statistical analyses given the many orders-of-magnitude difference in transformation efficiencies and the small biological error (e.g. PMID = 26422227, 30252595, 27041224).

Minor points

- The legend for Figure S9 should read "approach".
- The sentence on lines 191-2 is very confusing.
- Line 193: should read "FnCas12a"
- Line 210: "nuclease" should be plural.
- Line 88: "undergo" should be singular.

We thank the reviewer for catching these errors. We have addressed each instance in the text as recommended.

Reviewer #2 (Remarks to the Author):

All matters raised by me have been dealt with appropriately.

We thank the reviewer for their feedback.